# Song flight and 3D thermal detection provide evidence for bat attraction to wind turbines in Central Europe
Martina Nagy [1] ✉, Klaus Hochradel[2], Claudia Haushalter[1], Ralph Simon [3,4], Natalie Weber [5], Oliver Behr[6] & Mirjam Knörnschild [1,7,8]

Fatal interactions with wind turbines are a major threat to bat populations worldwide. Yet, the ultimate causes for bats colliding with wind turbines remain elusive. Using an extensive acoustic data set recorded at nacelle height in different parts of Germany, we show that feeding and social activity occur at all studied wind turbines. At least seven bat species (accounting for 95% of German bat fatalities) perform song flight at wind turbines, a behavior related to mating and courtship, indicating that males may find wind turbines attractive for establishing mating territories. Male songs broadcast over considerable distances and could function as acoustic beacons attracting females to turbine sites. Analysis of 3D thermal detection shows that bat density is higher in the rotor swept zone than in the free air space surrounding turbines. This strongly suggests that bats actively approach turbines, possibly in search of mating, roosting and/or foraging opportunities.

Wind energy plays a pivotal role in reducing $CO_2$ emissions and ultimately for mitigating climate change[1], but its negative impacts on wildlife remain an ongoing urgent concern (reviewed in ref. [2]) in view of the global biodiversity crisis. Wind turbines pose significant dangers to bats in multiple ways, including direct mortality, habitat loss, and displacement[3,4]. Fatal interactions (based on the assumption that bats colliding with the rotor are unlikely to survive, we use the terms collision, mortality, and fatality inter-changeably) with wind turbines have become one of the main sources of mortality for many bat species worldwide[5,6], some of which are under high legal protection (summarized in ref. [7]). The extensive numbers of bats being killed at wind turbines have raised serious concerns about the cumulative impacts on bat populations. Due to long generation times and low reproductive rates, high adult survival is essential for maintaining bat populations[8,9]. As yet, only operational mitigation (i.e., feathering the rotors of wind turbines during times of high collision risk for bats) has been shown to substantially reduce bat mortality at wind energy facilities (e.g. refs. [10,11]. but see refs. [12,13]). Knowing that high numbers of bat casualties in the temperate northern hemisphere have consistently been associated with low wind speeds and/or the migration and mating season, curtailment strategies currently mainly rely on easily measurable variables such as wind speed, time of year, time of night, and bat activity level[10] (but see ref. [11] for more complex variables like e.g., landscape features). In North America, for

example, loss in annual energy production has been estimated to be mostly below 1% and up to 3.2% for curtailment during the main collision risk periods of the year[13]. Similarly, a German study that reduced bat mortality from twelve to two dead bats per year with turbine-specific curtailment algorithms found that yearly revenue decreased by 1–2%[10]. Costs may be higher for modern turbines, but published data on this are lacking. To improve the efficiency of bat fatality mitigation and reduce associated costs, it could be very useful to understand not only the correlates of bat fatalities (e.g., wind speed and bat activity) but also the underlying causes of bat susceptibility to wind turbines.

The ultimate causes of bat collisions with wind turbines remain poorly understood and many proposed hypotheses have yet to be tested. Hypotheses include random or coincidental bat collisions with wind turbines as well as bat attraction to wind turbines[14–17]. Some common patterns of bat fatalities oppose the notion that bats randomly collide with wind turbines. In the temperate regions of Europe and North America, bat species most susceptible to wind turbines are typically tree-roosting bats ("tree bats")[14] and/or open space or edge space aerial insectivores that use quasi-constant frequency signals of high intensity to detect prey over larger distances[18,19]. Many of the affected species perform long-distance migrations during spring and autumn and fatality rates are highest during late summer and autumn with an occasional fatality peak in spring as well (reviewed in

[1]Museum für Naturkunde, Leibniz-Institute for Evolution and Biodiversity Science, Berlin, Germany. [2]Institute of Measurement and Sensor Technology, UMIT-Private University for Health Sciences, Medical Informatics and Technology GmbH, Hall in Tirol, Austria. [3]Nuremberg Zoo, Nuremberg, Germany. [4]CoSys-Lab, Antwerp University, Antwerp, Belgium. [5]Max Planck Institute of Animal Behavior, Radolfzell, Germany. [6]OekoFor GbR, Freiburg, Germany. [7]Evolutionary Ethology, Institute for Biology, Humboldt-Universität zu Berlin, Berlin, Germany. [8]Deutsche Fledermauswarte e.V., Berlin, Germany. ✉e-mail: martina.nagy@mfn.berlin

ref. 6). Yet, on a global scale the only apparent common pattern is that bats adapted to forage in open and edge spaces appear more prone to collisions with wind turbines than bats occupying other foraging niches[6].

Bats may be attracted to wind turbines when searching for feeding, roosting and/or mating opportunities (e.g. refs. 14,15,17,20). This could involve long-distance attraction, where bats are drawn to wind turbines from farther away, or increased local attractiveness, where wind turbines have traits that make these sites more appealing to bats already in the vicinity or commuting through the area. Indirect evidence for this comes from studies showing that preconstruction surveys of bat activity are poor predictors of post-construction bat activity levels and collision risk, possibly because the presence of turbines alters the bats' habitat and behavior (e.g. refs. 21,22.). It has been proposed that insects gather around turbines which would then serve as food sources for bats[14], but studies investigating the feeding-attraction hypothesis have produced mixed results. For example, using thermal cameras at wind turbines in the US, Horn et al.[23]. observed bats primarily foraging in and around the rotor-swept zone of a turbine and interpreted a significant correlation between insect and bat activity at turbines as evidence for bat attraction to insect patches. In contrast, one study from Canada did not find significant differences between the proportion of feeding buzzes (i.e., the characteristic increase of echolocation call emission rates during the final phase of prey capture[24]) at nacelle height, 30 m height and ground level and concluded that bats were not specifically attracted to feed at wind turbines[25].

Cryan[20] proposed that bat casualties at wind turbines are a consequence of mating behaviors that evolved in association with trees. According to this hypothesis, the highest trees in a landscape could serve as meeting points for tree bats during the mating season and the attraction effect of wind turbines results from their confusion with tall trees. It is plausible that both long-distance attraction—where bats are drawn to turbines from afar—and increased local attractiveness—where turbine sites may provide apparently favorable conditions for mating displays once bats are nearby—play a role. Behaviors of bats at wind turbines observed with thermal cameras in the US are well in accordance with such a scenario. The close approaches to turbines, flight maneuvers, hovering and chasing of other bats in the immediate vicinity of turbines, and the fact that bats appeared to actively orient toward turbines using air currents and vision are suggestive of behavioral adaptations for searching social partners at trees but also roosting sites or food[26].

In most Central European bat species, predominantly killed at wind turbines, males defend mating roosts and form transient harem groups with females during the mating period in late summer and autumn. Males of the noctule bat (*Nyctalus noctula*), Leisler's bat (*N. leisleri*), Nathusius' pipistrelle (*Pipistrellus nathusii)*, common pipistrelle (*P. pipistrellus*), and soprano pipistrelle (*P. pygmaeus*) use tree cavities as their mating roosts and attract females by singing from the roost entrance but also with conspicuous song flight[27–31]. In the parti-colored bat (*Vespertilio murinus*) song flight has been observed at high buildings, rock faces, and quarries[32]. For all these bat species, wind turbines might induce long-distance attraction if males perceive them as suitable for establishing mating territories and attracting females. Male song flight at turbines could further enhance the local attractiveness of these sites for females, reinforcing their use as mating locations. Wind turbines are visually highly conspicuous landmarks, and males are likely to profit from a large signaling range of their songs in the surrounding open habitats. Thus, as already suggested[20], the mating attraction hypothesis could be tested by investigating whether bats produce social vocalizations related to mating at wind turbines.

Here, we investigated whether bats are attracted to wind turbines using an extensive acoustic data set of 83,292 recordings with bat vocalizations recorded at nacelle height. We described the spatio-temporal distribution of feeding and social activity of bats (i.e., by investigating feeding buzzes and social vocalizations within bat recordings) to assess how frequent those behaviors occur across wind turbines and how they relate to the phenologies of the affected bat species. If bats are attracted to wind turbines for mating purposes, we predicted to find social vocalizations that are associated with song flight of male bats. In addition, we analyzed 30 h of stereo thermal

recordings of 3D thermal imaging of bats. If bats are attracted to and interact with turbine structures, we hypothesized that bat density should increase near the nacelle and within the rotor swept area of a turbine as compared to their density in free space surrounding the turbines. We find that multiple bat species, including those most frequently killed at wind turbines, regularly forage and perform mating-related song flight at turbines and 3D thermal imaging reveals the highest bat densities in the rotor-swept area near the nacelle. In summary, our results suggest that bats actively approach wind turbines, likely in the context of mating, roosting and/or foraging.

## Results

### Spatio-temporal distribution of feeding and social activity at wind turbines

Of 83,292 recordings with bat vocalizations, 1197 files contained feeding buzzes (1.4%), and 983 files contained social vocalizations (1.2%). Both the number of files with feeding buzzes and social vocalizations were positively correlated to the number of files with bat echolocation calls at a turbine (Supplementary Fig. 1, Spearman rank correlations on $n = 26$ turbine years; feeding buzzes: rho = 0.81, $P < 0.0001$, social vocalizations: rho = 0.65, $P = 0.0003$), indicating that the proportion of feeding and social activity was similar across the studied wind turbines.

In total, we identified 1598 individual feeding buzzes from six bat species (-groups) (Fig. 1A, Supplementary Fig. 2) and 4129 social vocalizations from ten bat species (-groups) (Fig. 1B, Supplementary Figs. 3–5, Supplementary Tables 1, 2). Feeding and social activity at wind turbines was found for *N. noctula*, *P. nathusii*, *P. pipistrellus*, and *P. pygmaeus*. In addition, we detected social vocalizations from *N. leisleri*, *V. murinus*, and *Plecotus spp*. It is probable that the feeding buzz recordings assigned to the Nyctaloid group also included feeding buzzes from *N. leisleri* and *V. murinus*, which could not be reliably identified due to overlapping echolocation frequency ranges of these species. Feeding and social activity were commonly present at all studied wind turbines and sites (Fig. 1).

Previous studies on the same wind turbines showed that bat activity peaked between the second half of July and the first half of September[10,33,34]. Similarly, we detected most feeding and social activity between July and September (Supplementary Fig. 6). For feeding activity, the temporal distribution was overall similar for Nyctaloid and Pipistrelloid bat species (Supplementary Fig. 7), whereas social vocalizations (social calls and songs) were also commonly found in October in all Pipistrelloid species but largely absent in Nyctaloids (Supplementary Figs. 8–10).

GLMMs revealed higher probabilities of feeding and social activity at wind turbines for Pipistrelloid bats compared to Nyctaloid species across all months, with stronger differences for social activity (log-odds difference = 1.88 for social activity vs. 0.93 for feeding activity, both $p < 0.001$, Table 1, Fig. 2, Supplementary Fig. 11). Significant interactions showed contrasting temporal patterns: Nyctaloid social activity at wind turbines tended to decline over the year, while Pipistrelloid activity tended to increase in late summer and autumn. For feeding activity, a significant cubic interaction term (log-odds difference = 1.37, $p < 0.01$) indicated complex temporal trends, with a significant difference between species in October (Supplementary Table 3). For social activity, post-hoc tests showed significant species group differences in both September and October, despite no significant month-to-month changes within species (Supplementary Table 4). This suggests that Pipistrelloid species exhibit higher social and feeding activity at wind turbines compared to Nyctaloid species, with the differences in social activity being more pronounced, particularly in late summer and autumn.

### Bat song at wind turbines

Social vocalizations were further classified into social calls and songs. Social calls are typically simple, short, and discrete vocalizations that are produced irregularly or singly and may serve as contact calls for instance. Songs consist of different elements (i.e., discrete acoustic units) combined into often stereotypic sequences and accompany behaviors related to courtship and territoriality[35–37]. Thus, bat song at wind turbines is of special interest as

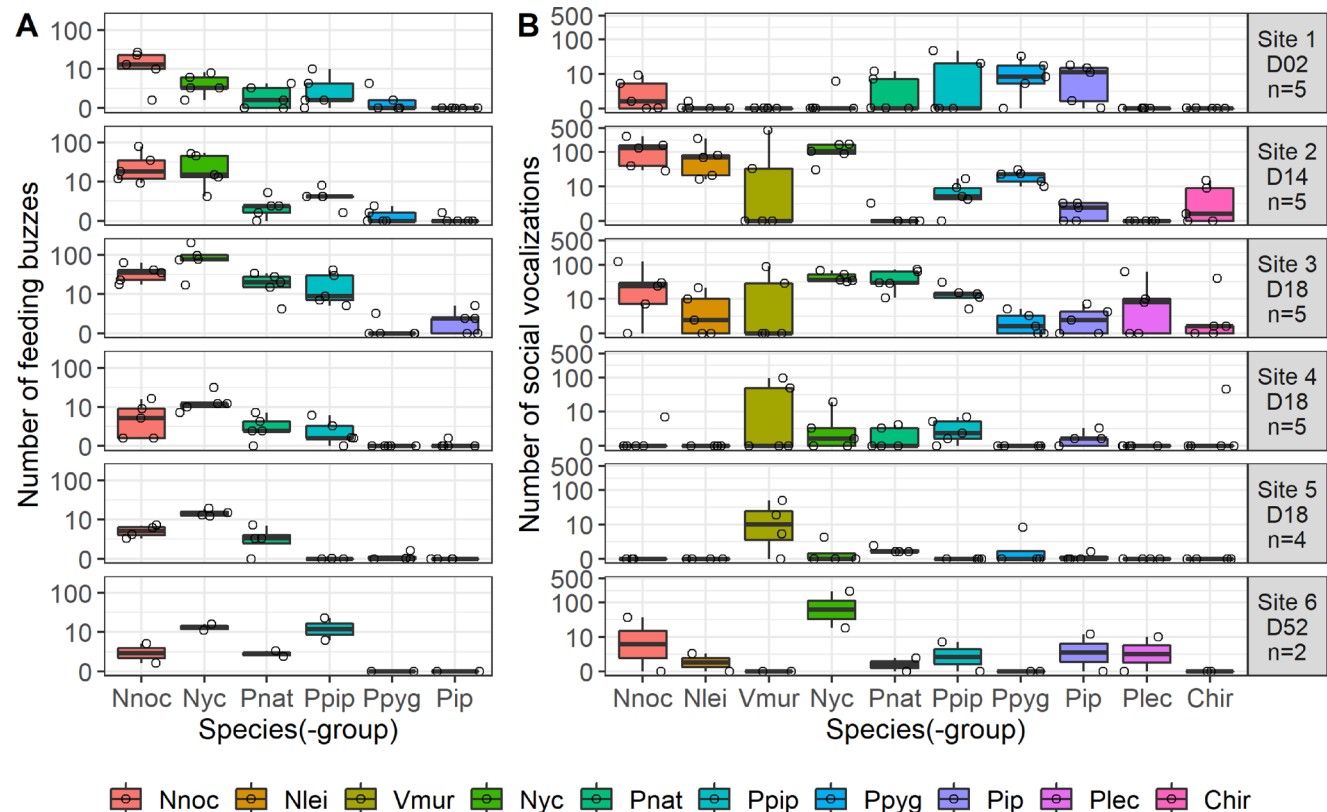

**Fig. 1 | Occurrence of feeding buzzes and social vocalizations of bats at wind turbines.** Boxplots show the median number of recorded **A** feeding buzzes and **B** social vocalizations (the sum of social calls and song elements) per species (-group) and site. Whiskers extend to 1.5x the interquartile range (IQR). Open circles depict the original data per species (-group) and turbine year. Panel labels report site numbers, the natural regions of Germany where wind turbine sites were located, and the number of turbine years sampled per site (*n* = 5, 5, 5, 5, 4, and 2; total *n* = 26 wind turbine-years from six biologically independent wind turbine sites). Panels have a pseudo-logarithmic y-axis. Species (-group) abbreviations: Nnoc, *N. noctula*; Nlei, *N. leisleri*; Vmur, *V. murinus*; Nyc, *Nyctaloid*; Pnat, *P. nathusii*; Ppip, *P. pipistrellus*; Ppyg, *P. pygmaeus*; Pip, *Pipistrellus spp.*; Plec, *Plecotus spp.*; Chir, Chiroptera.

**Table 1 | Results of GLMMs testing the effect of species group (Pipistrelloid and Nyctaloid) and month on feeding and social activity of bats at wind turbines**

| | Feeding buzzes | | | | Social vocalizations | | | |
|---|---|---|---|---|---|---|---|---|
| **Predictor** | **Estimate** | **Std. Error** | **z-value** | **p-value** | **Estimate** | **Std. Error** | **z-value** | **p-value** |
| (Intercept) | −4.85 | 0.14 | −35.4 | **<0.001** | −6.65 | 0.33 | −19.93 | **<0.001** |
| Species Group (Pip) | 0.93 | 0.17 | 5.41 | **<0.001** | 1.88 | 0.35 | 5.42 | **<0.001** |
| Month (Linear) | −0.51 | 0.36 | −1.42 | 0.16 | −1.16 | 0.87 | −1.34 | 0.18 |
| Month (Quadratic) | −0.74 | 0.32 | −2.3 | **0.02** | −2.58 | 0.80 | −3.22 | **<0.01** |
| Month (Cubic) | −1.19 | 0.30 | −4.02 | **<0.001** | 0.11 | 0.64 | 0.17 | 0.86 |
| Month (Quartic) | 0.06 | 0.26 | 0.22 | 0.83 | 0.24 | 0.49 | 0.48 | 0.63 |
| Month (Quintic) | −0.68 | 0.19 | −3.51 | **<0.001** | −0.15 | 0.37 | −0.40 | 0.69 |
| Species Group (Pip) x Month (Linear) | 0.16 | 0.49 | 0.32 | 0.75 | 3.00 | 1.07 | 2.82 | **<0.01** |
| Species Group (Pip) x Month (Quadratic) | 0.98 | 0.45 | 2.18 | **0.03** | 2.52 | 0.96 | 2.62 | **<0.01** |
| Species Group (Pip) x Month (Cubic) | 1.37 | 0.43 | 3.18 | **<0.01** | −0.08 | 0.86 | −0.09 | 0.93 |
| Species Group (Pip) x Month (Quartic) | −0.26 | 0.39 | −0.66 | 0.51 | 0.05 | 0.72 | 0.06 | 0.95 |
| Species Group (Pip) x Month (Quintic) | 0.09 | 0.32 | 0.27 | 0.79 | 0.32 | 0.56 | 0.58 | 0.56 |

Significant *p*-values (<0.05) are highlighted in bold.

singing males may attract females to those sites (e.g. refs. 28,38,39). Of the 4129 social vocalizations recorded at wind turbines, 2811 were classified as social calls (Supplementary Fig. 3, Supplementary Fig. 12, Supplementary Tables 1, 2) and 1318 were song elements (Fig. 3, Supplementary Tables 1, 2). We defined song elements that were uttered as part of temporally

successive sequences as song events. We identified 92 individual song events with a mean duration of 17.5 ± 20.3 s.

Seven bat species were found to sing at wind turbines: *N. noctula, N. leisleri, P. nathusii, P. pipistrellus, P. pygmaeus, Plecotus spp.*, and *V. murinus*. These species account for 95% of the documented bat fatalities at wind

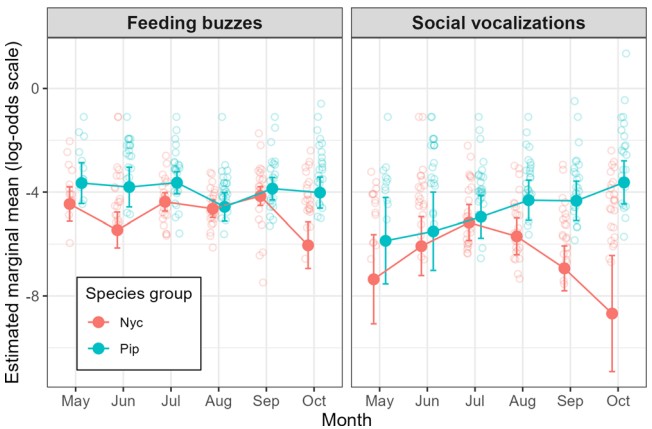

**Fig. 2 | Marginal means of feeding and social activity at wind turbines across months for Nyctaloid and Pipistrelloid species groups.** The graphs depict the estimated marginal means (± 95% confidence intervals) for feeding buzzes (left panel) and social vocalizations (right panel) from binomial GLMMs. The models tested the effect of species group and month on the proportion of Nyctaloid or Pipistrelloid call recordings that contained feeding buzzes or social vocalizations (social calls and/or songs) of the respective species group. Within each month, Nyctaloid (Nyc) and Pipistrelloid (Pip) data points are shown with a slight horizontal offset on the x-axis for better visibility. Lines connecting the points are included to aid in visualizing trends across months but do not imply a linear or smooth transition between months. Semi-transparent circles show the raw turbine-year data (log-odds of the proportion of call recordings containing feeding buzzes and social vocalizations), plotted on the same scale as the model estimates to illustrate the underlying data distribution. Apparent discrepancies between individual data points and marginal means reflect the binomial weighting of proportions: months in which echolocation activity remains high but social calls or feeding buzzes are nearly absent (e.g., October for Nyctaloids) exert stronger influence on the model estimates than months in which high echolocation activity coincides with consistent social or feeding activity. n = 26 biologically independent wind turbine years from 6 wind turbine sites, with site included as a random effect in the GLMMs.

turbines in Germany[40]. The two bat species most frequently killed at wind turbines in Germany (*N. noctula*: 34% of fatalities, *P. nathusii*: 30% of fatalities[40]) were also responsible for the majority of recorded song events: 41% *N. noctula* (38 song events) and 24% *P. nathusii* (22 song events). We found singing bats at all studied wind turbine sites and in 23 of 26 (88%) wind turbine–years (Fig. 3). On average 1.7 ± 0.9 species (-groups) were found singing per wind turbine-year (Range: 0-3 species (-groups)).

Singing bats were recorded in all months from May to October, and most singing activity occurred from July through October (Supplementary Fig. 6). Only *Pipistrellus* and *Plecotus* species were singing at wind turbines during October. Neither *N. noctula*, nor *N. leisleri* or *V. murinus* were found singing at wind turbines in late autumn. This coincides with the main period of male singing activity and the presumed mating period of these species, except for *V. murinus* for which singing males have only been reported from October through December in Germany (see Supplementary Table 5 for a summary of species characteristics). One possible reason may be that the mating season of *V. murinus* starts earlier than currently known, and/or that male song detected prior to the presumed mating season reflects territorial behavior of males attempting to establish mating territories.

### Covered flight distance and acoustic signaling range of singing bats
Mean song event duration at wind turbines was 17.5 ± 20.3 s with song events lasting between two and 98 s (Fig. 4A). Commuting flight speeds of the singing bats in our study range between 2.5 m/s (*Plecotus spp*[41].) and 6.0 m/s (*N. noctula*[42]) and result in an average distance covered during song flight of 97.1 ± 111.1 m (Range: 5–539 m, Fig. 4B, Supplementary Table 6). The calculated mean detection distance of the sound recording device (operated with a trigger threshold level of 37 dB SPL) at 20 °C and 60%

humidity for species specific peak frequencies of song (14.0–26.4 kHz) and species-specific call intensities (72–108 dB peSPL at 1 m) ranged between 25.1 m for the song of *Plecotus* spp. and 114.3 m for *V. murinus* song (see "Material and Methods" and Supplementary Tables 7 and 8 for details). For subsequent analyses, we used the species-specific detection distances of the recording device for bat song as a conservative proxy for the maximum distance a singing bat could cover without leaving the detection range of the acoustic recorder. In 37 of 92 song flights (40.2%), the songs lasted long enough for bats to cover distances that exceeded the respective species-specific song detection range, suggesting prolonged presence of singing bats in the rotor swept area, possibly because they were circling the turbine tower (Fig. 4D). The observation that song elements of those longer song flights alternately became fainter and louder at regular intervals (Supplementary Fig. 5) is also in line with the supposition of circling bats.

Based on species-specific call intensities and assuming a detection threshold of 20 dB SPL, humidity of 60% and a temperature of 20 °C, the resulting distances over which bat songs may be heard by conspecifics in open habitat ranged between 42 m for *Plecotus spp*. and 100 m for *P. nathusii* and *V. murinus*. The active space of bat songs is, however, greatly extended by song flight behavior such as circling of the turbine towers (Fig. 4C, Supplementary Table 9).

### 3D thermal imaging of bat density in the rotor swept area
Using stereo-thermal recordings of bat flight paths from six nights at four wind turbines, we detected 4468 and 1489 bat positions within a radius of, respectively, less than 60 m and 30 m from the nacelle (i.e., the shortest distance between the flight path and the center of the nacelle). We estimated bat density for hemispherical shells around the center of the nacelle with a thickness of 3 m each (Fig. 5A, B). Starting from a radius of about 10 m, bat density decreased exponentially with increasing distance to the nacelle and approached a constant level that probably reflects the distribution of bats in free air space (Fig. 5C). The higher bat density within the rotor-swept area compared to the surrounding area suggests that bats are attracted to wind turbines and actively approach them.

### Discussion
Our analysis of social vocalizations recorded at the nacelles of wind turbines in Germany reveals that at least seven European bat species (*N. leisleri*, *N. noctula*, *P. nathusii*, *P. pipistrellus*, *P. pygmaeus*, *Plecotus* spp., and *V. murinus*) perform song flight—a behavior that is related to courtship and mating—in the close vicinity of turbines including the rotor-swept zone. In accordance with a relation to mating behavior, we detected singing activity of bats to peak in late summer and autumn during the mating season. Bat song was documented at all studied sites and in 88% of the 26 wind turbine-years, suggesting that while it is not very common (92 song events in total), it is a consistently observed and recurrent behavior at wind turbines in Germany. The seven bat species for which we recorded songs in the immediate vicinity of turbines closely mirror the fatality patterns at wind turbines in Germany, accounting for 95% of reported bat fatalities. Moreover, the two bat species with the highest collision risk in Germany (*N. noctula* and *P. nathusii*: 64% of fatalities[40]) were also the most frequent singers (65% of song events). Thus, the occurrence of mating-related behaviors at wind turbines by the bat species most impacted by collisions supports the hypothesis that these bats perceive wind turbines as attractive structures for seeking mating opportunities, possibly because they try to establish mating territories and to attract females at the most visually conspicuous, tall structures in a landscape[20].

Because the sexes are segregated throughout most of the year in temperate bats, male mating strategies that increase the chances of meeting female mating partners are likely to play an important fitness role. During late summer and autumn, males of the bat species with the highest turbine-related mortality in Europe (i.e., *N. leisleri*, *N. noctula*, *P. nathusii*, *P. pipistrellus*, *P. pygmaeus*, and *V. murinus*) are known to establish mating roosts (in e.g., tree cavities or building crevices) in regions where females are expected to pass when migrating from nursery roosts to their winter roosts

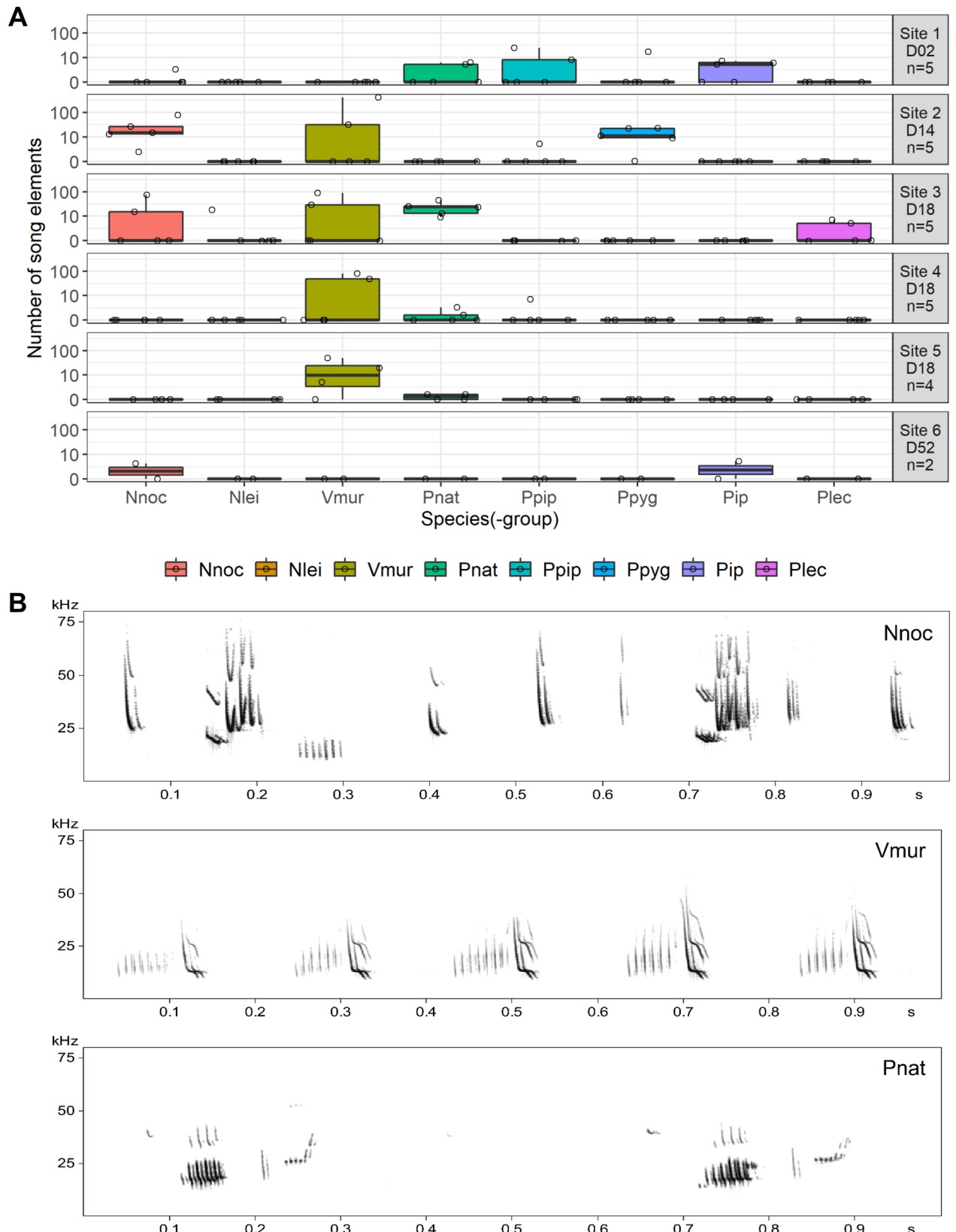

**Fig. 3 | Occurrence of bat song at wind turbines. A** Boxplots show the median number of recorded song elements per species (-group) and site. Whiskers extend to 1.5x the interquartile range (IQR). Open circles depict the original data per species (-group) and turbine year. Panel labels report site numbers, the natural regions in Germany where wind turbine sites were located and the number of sampled turbine years per site ((n = 5, 5, 5, 5, 4, and 2; total n = 26 wind turbine-years from six biologically independent wind turbine sites, see also Supplementary Fig. 12). Panels have a pseudo-logarithmic y-axis. **B** Song excerpts (1 s) from *Nyctalus noctula* (two song motifs), *Vespertilio murinus* (five song motifs), and *Pipistrellus nathusii* (two song motifs). Please note that in some cases, echoes of the songs are visible in the spectrograms. Species (-group) abbreviations: Nnoc *N. noctula*, Nlei *N. leisleri*, Vmur *V. murinus*, Pnat *P. nathusii*, Ppip *P. pipistrellus*, Ppyg *P. pygmaeus*, Pip *Pipistrellus spp.*, Plec *Plecotus spp.*

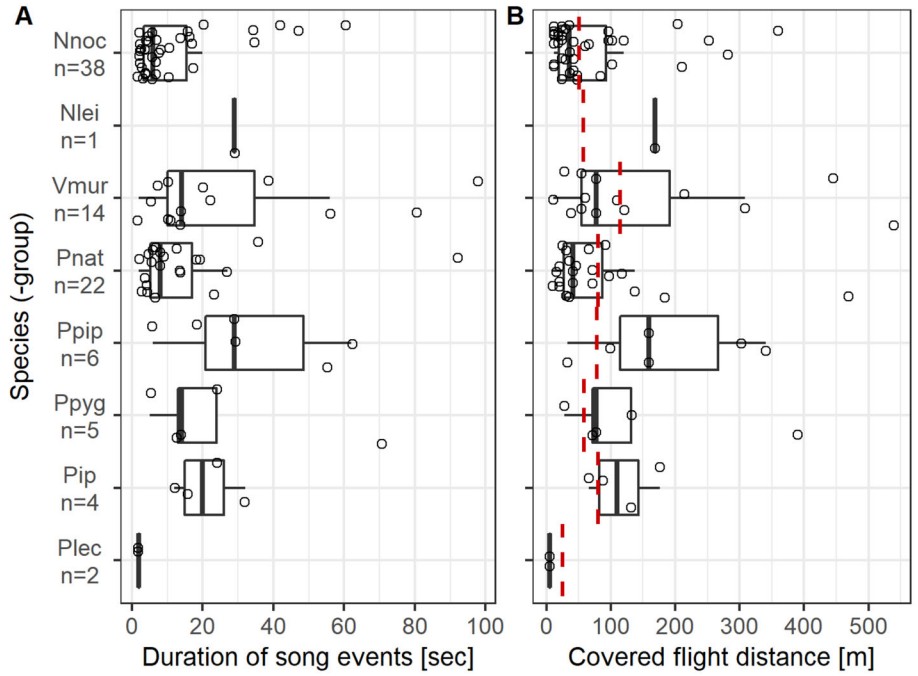

| Singing bat species | Active space [m] |
|---|---|
| *N. noctula* | 69.4 |
| *N. leisleri* | 81.4 |
| *V. murinus* | 100.2 |
| *P. nathusii* | 100.2 |
| *P. pipistrellus* | 96.6 |
| *P. pygmaeus* | 82.4 |
| *Plecotus spp.* | 41.8 |

Estimated signaling ranges at 20°C and 60% humidity over which songs can be detected by conspecifics in open space.

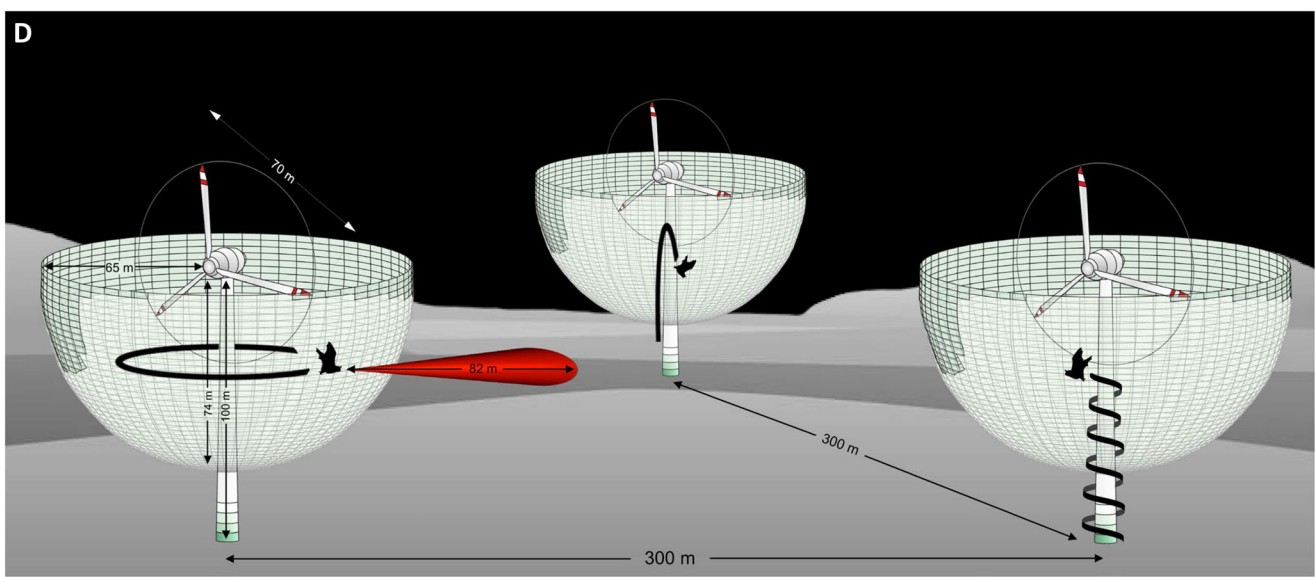

**Fig. 4 | Duration, covered flight distance, and active space of bat songs. A** Boxplots showing the distribution of song event durations across species (-groups). **B** Boxplots showing the distribution of flight distances bats covered per song event. The dashed red line at the species-specific detection range of the recording device for bat song (25.1–114.3 m, Supplementary Table 7) marks the covered flight distance after which a straight-ahead flying bat should have left the detection range of the recording device installed at the turbine nacelle. Thus, covered flight distances larger than the species-specific song detection range of the recorder suggest singing bats were circling around the turbine tower. Open circles in (**A**) and (**B**) depict raw data and whiskers extend to 1.5× the interquartile range (IQR). Open circles represent independent song events (*n* = 92 in total; species-specific n values are indicated on the y-axis). **C** Active space of songs across species, i.e., estimated acoustic signaling ranges over which bat songs are broadcasted. **D** Schematic diagram of a wind turbine site with 70 m turbine diameters, 100 m nacelle heights and 300 m average distances between turbines showing the detection range of the recording device installed in the turbine nacelle for calls with frequencies of 20 kHz (Supplementary Table 8) and bat song flight around the turbine tower. The active space of bat songs is illustrated as a sound beam of 82 m (i.e., the mean across the bat species in **C**; range: 42 m for *Plecotus spp.*, 100 m for *V. murinus*). Beam width is depicted arbitrarily. The acoustic shadow of the tower is not shown.

as well as adjacent to nursery and winter roosts[27,43]. The conspicuous songs produced by males have been shown to attract conspecifics in a playback experiment[44] and for instance in *P. pygmaeus* harem size was largest in the male that spent most time in song flight display[28], suggesting that the function of male song is to attract females to the mating roost. Our calculations show that the songs of bat species singing at wind turbines are broadcasted over much higher distances than echolocation calls (42–100 m) in the surrounding open habitat of wind turbines and may thus function as acoustic beacons which permit females to localize male mating territories;

male song has been experimentally demonstrated to elicit approach behavior in females of a neotropical bat[38]. The large distances covered during many of the song flight events (40%) we recorded at wind turbines and the regular nature by which individual song elements of a song became louder and fainter speak against bats that are singing while in transit but are suggestive of bats, for example, circling or flying up and down the turbine tower (Fig. 4D). The presumed song flight trajectories recorded at wind turbines resemble the reported patrolling of the territory around the mating roost during song flight display, which for instance in *P. pipistrellus* and *P.*

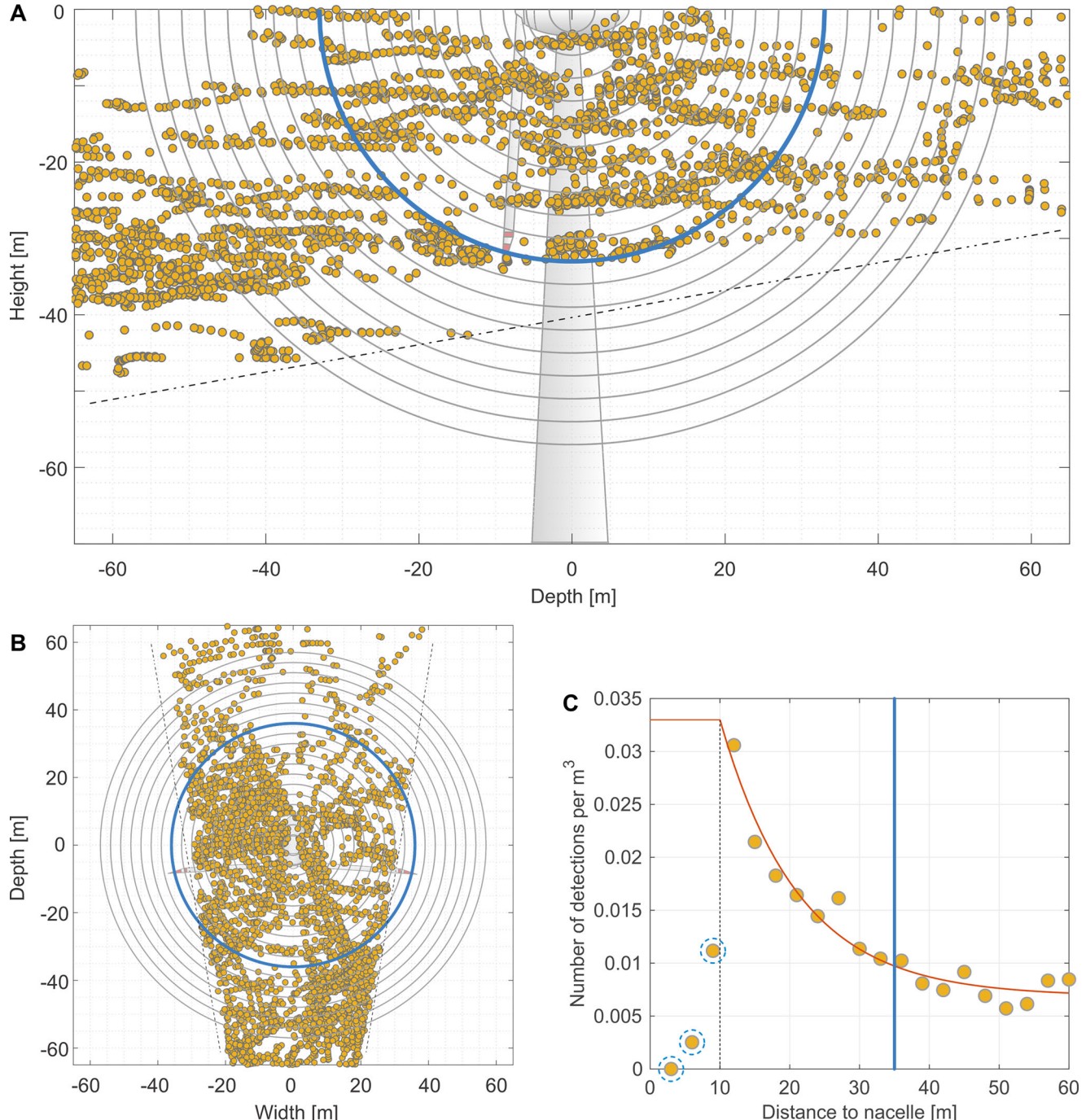

**Fig. 5 | Bat density decreases with increasing distance to the nacelle. A** Lateral and **B** top view of bat positions recorded for two independent nights ($n = 2$) in 2008, shown as representative examples, at one study turbine with a rotor radius of 35 m (the limit of the rotor-swept zone is highlighted in blue). All recorded bat positions within a radius of 60 m around the center of the nacelle are shown. Concentric circles around the nacelle represent hemispherical shells (3 m thickness each) used to calculate bat density in the rotor swept area and its open space surrounding area as shown in (**C**). The dotted lines show the limits of the field of view of thermal cameras.

**C** Bat density (number of bat detections per m³) per hemispherical shell around the center of the nacelle, calculated from six independent nights ($n = 6$) recorded at four wind turbines from two sites (total 29.9 h of stereo-thermal recordings). The vertical blue line indicates the limit of the collision-risk zone. The nacelle occupies a large part of the volume in close vicinity to the nacelle (<10 m, blue dashed borders around data points) explaining the low activity in this area. The volume of hemispherical shells initially increased with increasing distance from the nacelle and then decreased due to a decreasing field of view of the thermal cameras (**A**).

*pygmaeus* often depicts elliptic or circular flight paths and includes regular routes that often follow linear structures such as building walls or the edges of woodland[27,28]. Patrolling behavior during song flight should further increase the signaling range of male song and female encounter probability. Thus, the fact that we detected male song flight at wind turbines almost certainly contributes to increasing the local attractiveness of these sites for females[38], regardless of whether wind turbines have any features that attract

male bats to those sites in the first place. This mechanism may help explain the observed female-biased mortality at wind turbines during migration that has been reported for long-distance migratory bat species such as *N. noctula* and *P. nathusii* (e.g. refs. [45,46]).

Because we do not have data on whether male bats were using the studied sites for courtship before the construction of the turbines, it is difficult to conclusively say whether wind turbines attract male bats who

search for mating opportunities. However, based on what we know about male courtship behavior of the bat species concerned, it appears unlikely that the mostly open farmland was suitable for advertising males prior to the construction of turbines. Courtship behavior of those bat species is tightly linked to structures, first because trees or buildings are necessary for establishing mating roosts and, second because song flight is performed in close proximity to mating roosts and flight paths also often follow linear or prominent landscape features such as forest edges or buildings[27,28,43,47]. Therefore, it appears plausible that the construction of conspicuous wind turbine sites in a homogeneous and structurally poor environment adds apparently suitable sites for male courtship. Moreover, for instance in *P. nathusii*, mating roosts and song flight territories have been reported to be often spatially clumped and male neighbors tended to synchronize their vocalization activity[48], which may create a conspicuous chorus that is more easily detected by females from afar. Thus, chorusing males at wind turbine sites could potentially achieve a considerable signaling range given the open surroundings and function as acoustic beacons to facilitate their detection by females (Fig. 4D).

To date, there is evidence of song produced in flight in only one (*Lasionycteris noctivagans*) of the three North American migratory bat species (*Lasiurus cinereus*, *L. borealis*, *L. noctivagans*) most affected by wind turbines[49]. However, recent studies indicate that *L. cinereus* uses and is attracted to social calls during migration[50,51]. Additionally, in one study most bats of these species found dead below wind turbines exhibited physiological signs of mating readiness[52], suggesting that social and mating related behaviors may contribute to their attraction to wind turbines[20]. This underscores the importance of further research into the mating systems and behaviors of these North American species, including mating-related social communication and the potential use of song flight to attract mating partners.

Our results on bat density at nacelle height are strong evidence for bat attraction to wind turbines (see also e.g. ref. [53]). Based on the 3D thermal imaging data set on bat density at wind turbines it is now possible to draw quantitative conclusions about the exact spatial behavior of bats in the rotor swept area of wind turbines. While our dataset is limited in size, it provides the best available evidence for bat attraction to wind turbines at nacelle height. Future studies with larger sample sizes will be essential to further validate these findings. We were able to show that bat density increased with increasing proximity to the nacelle and was clearly higher than in free air space surrounding the turbine. Similarly, in a paired design study from Britain, recorded activity of *P. pipistrellus* was 37% higher at turbine sites than at open habitat controls consistent with *P. pipistrellus* attracted to turbines; yet, species-specific differences appeared to exist given that *P. pygmaeus* activity was similar between turbines and open habitat controls[54]. Bat behaviors that have been observed at turbines using 2D thermal videos include bats actively foraging around and within the rotor-swept zone, investigating turbine blades and orienting toward wind turbines using air currents and vision. The authors suggested that bats may be attracted to patches of insects at wind turbines and may view them as potential roost trees and/or mating sites[23,26]. Our results show that foraging activity occurred at all studied wind turbines with feeding buzzes from 2 to 6 species (-groups) detected per wind turbine year and may provide additional support for the feeding attraction hypothesis. In contrast, because feeding buzz rates did not differ between ground level, 30 m height of met towers and the height of the nacelle (67 m) at wind turbine sites in Canada, the authors concluded that bats were not specifically attracted to wind turbines for feeding[25]. However, these results might not necessarily indicate that feeding rates of bats at the height of the nacelle were similar before the construction of wind turbines, because bat species composition and activity are known to change with increasing altitude (e.g. refs. [55–57]), and because turbine construction probably alters the bats' behaviors in the area[21,58]. It might therefore be worthwhile to compare feeding rates recorded at the nacelle of wind turbines with those recorded in open habitats of the same height before turbine construction, to gain data on feeding rates at nacelle height before and after the construction of wind turbines (e.g., feeding rates could be recorded at wind masts before turbine construction[59]).

One important question is what the evidence for bat attraction to wind turbines for mating and feeding purposes might mean in relation to existing mitigation strategies such as feathering wind turbines during times of high mortality risk for bats. We found a strong positive correlation between recorded feeding and social activity (i.e., the number of feeding buzzes and social vocalizations) and overall bat activity at the surveyed turbines, which suggests that measuring overall bat activity should in general reliably reflect the level of social and feeding activity at a turbine. An interesting finding from the GLMMs was the significantly elevated feeding and social activity of Pipistrelloid bats as compared to Nyctaloid bats, particularly in late summer and autumn. The difference in social activity was more pronounced between the species groups, suggesting that Pipistrelloid bats are not only more socially active at wind turbines than Nyctaloid bats but also that these differences become more marked as the year progresses and, thus, during the migration and mating season of these species. The particularly large differences in social activity between Nyctaloid and Pipistrelloid species groups at wind energy facilities in September and October could potentially be explained by differences in migration timing, as well as by the fact that two species within the Pipistrelloid group are more sedentary (e.g. refs. [60], [61]). One likely cause of this result is the decreasing detection range of acoustic recorders for bat calls of increasing frequency. Whereas Nyctaloid echolocation calls are lower in frequency (16–30 kHz) than Pipistrelloid echolocation calls (37–60 kHz), the songs of these bat species are in a similar frequency range (peak frequencies of 13–26 kHz). Therefore, assuming similar echolocation (and feeding) activity, acoustic detectors will record fewer echolocation calls (and feeding buzzes) from Pipistrelloid than from Nyctaloid species, whereas the number of song recordings is not expected to differ, assuming similar social activity of the two bat species groups. In addition, it is possible that all or some Pipistrelloid species tend to be more socially active at wind turbines than Nyctaloid species. A study by Bach et al.[62] may support this assumption. The authors report that fatality numbers are higher than expected given the relatively low measured acoustic activity for *P. nathusii* in coastal areas of northwestern Germany, which include important reproductive regions and migration routes of this species. Fatalities peaked during the mating season in August and September and were highly female-biased (67%), in contrast to a rather unbiased sex-ratio of fatalities across Germany[40,63]. It would be highly interesting to examine those recordings for male songs of *P. nathusii*, as female attraction by advertising males that position themselves at wind turbines in the migration routes might help explain the highly female-biased fatality ratios. The higher ratio of fatalities to number of recorded calls in *P. nathusii* as compared to other wind turbine susceptible bat species is a known phenomenon in Germany, especially at coastal sites, and has been accounted for in a recent update (since version 6.1) of the so-called ProBat Tool[64] for calculating turbine-specific curtailment algorithms[65].

There are hardly any data on the long-term temporal variability of bat activity at wind turbines. To date, there is only one study from Germany on a large acoustic dataset from 5 years, which found that variance in bat activity in different years was higher than between turbines of the same site and comparable to the variance between different sites. However, the dataset did not permit conclusions on whether bat activity at a wind turbine increased or decreased over longer periods[66]. Attraction of bats to wind turbines in search of mating and foraging opportunities may lead to increased mortality, which, over time, may be reflected in decreased acoustic activity, turning wind turbine sites into ecological traps for these species (i.e., habitats that attract organisms but reduce their survival or reproduction; e.g., refs. [67,68]). However, it is also possible that acoustic activity at turbines may initially increase during the first years following turbine construction if local and/or migrating bats need some time to discover those new sites. In Central Europe, acoustic surveys that inform curtailment algorithms are typically conducted over the first two years post-construction and wind turbines are subsequently operated with these algorithms during the 20 years of their approved runtime. The acoustic dataset of this paper has been recorded more than 6 years after the construction of the sampled wind turbines, and we can thus not assess whether there have been changes in the level of social

**Table 2 | Sample size of the acoustic dataset from wind turbines**

| Site | Number of sampled turbines/turbine years | Years | Number of sampled hours | Number of sampled nights | Number of bat recordings |
|------|------------------------------------------|-------|-------------------------|--------------------------|--------------------------|
| 1 | 4/5 | 2012, 2014 | 10,784 | 803 | 5644 |
| 2 | 4/5 | 2012, 2015 | 9313 | 885 | 27,055 |
| 3 | 4/5 | 2012, 2014 | 11,019 | 891 | 35,745 |
| 4 | 4/5 | 2012, 2014 | 11,078 | 869 | 5715 |
| 5 | 4/4 | 2012, 2014 | 10,107 | 755 | 5247 |
| 6 | 2/2 | 2012 | 2417 | 228 | 3886 |
| Total | 22/26 | | 54,718 | 4431 | 83,292 |

and foraging activity compared to the first turbine operation years. If further studies find that bat activity levels change over time at wind turbines, then additional acoustic monitoring data recorded several years after turbine construction may be necessary to reevaluate bat activity levels and tailor curtailment algorithms accordingly. In summary, we believe that only well-parametrized curtailment (e.g., refs. 10,11,69), incorporating stricter curtailment during the mating period, will effectively prevent bats from being attracted to an ecological trap.

## Methods
### Acoustic data set
Acoustic recordings were sampled as part of RENEBAT II and III. These research projects developed methods to reduce the collision risk of bats with wind turbines in Germany[70,71]. Data used in this paper were recorded in 2012, 2014, and 2015 at 22 wind turbines (2–4 turbines per site), six sites and in five natural regions of Germany. At four sites, one turbine was sampled in 2 years resulting in 26 turbine years of acoustic recordings (Table 2, Supplementary Fig. 13). The RENEBAT II and III projects recorded acoustic data at wind turbines with several acoustic detector systems; for the present study, we included only recordings obtained with the Avisoft Bioacoustics system (Glienike, Germany), which provided the microphone sensitivity and signal quality required to reliably identify feeding buzzes and social vocalizations[70,71].

All wind turbines were from ENERCON GmbH (Aurich, Germany) of the types E66 and E70 with rotor diameters of 66 m and 70 m and nacelle heights between 63 m and 100 m. Enercon service teams installed detector systems (Avisoft Bioacoustics, Glienike, Germany) in the nacelle of each turbine to continuously record bat activity in the rotor swept area of wind turbines. Holes were drilled in the nacelle floor to position microphones inside and in the bottom of the nacelle between the rotor and the tower with the microphone pointing downwards, through the nacelle floor. Detectors were installed between April and beginning of July and were run until September to December, covering the main activity period of bats from July to September at all turbines.

The Avisoft system consisted of the USG 116Hnbm (Avisoft Bioacoustics, Glienike, Germany) and an aluminum microphone disc connected to an industrial PC running the RECORDER software (Avisoft Bioacoustics, Glienike, Germany). The microphone disc contained an electret microphone (FG23629-P16, Knowles, Itasca, Illinois, United States), a test signal generator (Piezoelectric speaker KPUS-40FS-18T-447) and a microphone heating (i.e., to reduce humidity and associated temporal changes in microphone sensitivity). Every afternoon, the test signal generator produced a frequency-modulated signal between 30 kHz and 50 kHz. The recording of the test signal was used for measuring the daily sensitivity of detector systems for ultimately determining valid recording periods. The RECORDER software was run with the following settings: 300 kHz sampling rate, 16-bit Format, 37 dB SPL trigger level, 0.3 s pre-trigger, 1 s post-trigger and nightly timer from 15:30 to 08:00 (UTC + 02:00). Additionally, the bat call filter was enabled with the following trigger settings: FM min sweep rate -9, FM max sweep rate -1, FM min duration 1 ms, CF min sweep rate -2, CF max sweep rate 1, CF min duration 2 ms, frequency range 15–80 kHz. Detectors ran continuously and produced valid data during 94% of the

nights sampled (detector downtimes were caused by microphone sensitivity outside a range of ±6 dB of the original calibration or by power or other microphone failures or other technical problems). The mean number of nights with valid data for 26 turbine years was 170 ± 40 (Range 76–210 nights) of a total of 181 ± 39 nights sampled per turbine year (Range 112–228 nights). This resulted in 107,950 acoustic recordings of which 83,292 contained bat vocalizations sampled over 4431 nights and 54,718 h (Table 2).

### Acoustic analysis
**Identification of species groups**. We used the RECORDER software with the "RENEBAT settings" to automatically identify bat calls in the Avisoft recordings and to identify species groups. Using the "RENEBAT settings", the software first identifies bat calls with a bat call filter (*1. Configuration:* Pretrigger = 0.3 s, Hold = 1 s, Duration >0 s, Syllable > 0 s, Trigger event Level = 0.01%, Trigger Event Range = 15–100 kHz. *2. Bat Call Trigger Filter Settings:* FM min sweep rate = -9, FM max sweep rate = −1, FM min duration = 1 ms, FM max duration = 30 ms, CF min sweep rate = −2, CF max sweep rate = 1, CF min duration = 2 ms, CF max duration = 30 ms, FFT size = 512, Overlap = 87.5, Window = FlatTop, frequency range = 15–100 kHz, High/low frequency magnitude ratio > 6 dB, fc = 15 kHz, magnitude threshold = −80 dBFS, hold time = 2 ms). Subsequently and based mainly on the end frequency of echolocation calls the software classified two species groups. The Nyctaloid group with end frequencies ≥8 kHz and <33 kHz contained the bat species *N. noctula*, *N. leisleri*, *E. serotinus*, *E. nilssonii* and *V. murinus*. *Myotis* as well as *Plecotus* species are classified as part of the Nyctaloid group, though these species are rarely recorded at nacelle height of wind turbines (<0.1% of bat recordings in refs. 33,59). Bat species with end frequencies ≥33 kHz and <65 kHz were classified to the Pipistrelloid group and contained the species *P. nathusii*, *P. pipistrellus*, and *P. pygmaeus*. *Barbastella barbastellus* is classified as part of the Pipistrelloid group, though this species was rarely recorded at nacelle height (<0.1% of bat recordings in refs. 33,59). All recordings that were automatically identified as bat calls by the software were subsequently manually checked, and misclassified noise or test signal recordings were removed from the dataset. Subsequently, we manually identified the bat species (-groups) that produced feeding buzzes or social vocalizations.

### Acoustic analysis of feeding buzzes
All recordings with bat calls (83,292) were manually reexamined for the presence of feeding buzzes and social vocalizations (see below). We used the software Avisoft SASLab Pro v5.2.13 and v5.2.14 (Avisoft Bioacoustics, Glienike, Germany) for sound analysis and spectrograms were generated using a Hamming window (1024 FFT, 75% overlap, 488 Hz frequency resolution, 0.512 ms time resolution).

When bats are homing in on flying insect prey, they increase echolocation call emission rates to obtain more information on the relative position of their target. Following prey detection, bats switch from the search phase to the approach phase, which is characterized by a reduction in signal duration and pulse interval and ends in a distinct terminal buzz. During the latter, repetition rates typically increase from approximately 100 calls/s up to more

than 200 calls/s in some species (e.g. refs. 19,24,72). We classified a call sequence as a feeding buzz when repetition rates ≥100 calls/s (i.e., pulse intervals ≤10 ms) indicated that the bat had reached the terminal buzz phase (Supplementary Fig. 2). We identified 1598 feeding buzzes in 1196 files. On average, these 1196 files contained 1.3 ± 0.8 feeding buzzes ranging from one to nine feeding buzzes per file. Furthermore, we manually identified the bat species (-groups) that had uttered the identified feeding buzzes based on call structure and frequency parameters of at least two search flight calls preceding the feeding buzzes. Species identification followed a custom-made identification key and data on echolocation calls from the literature (e.g[73–75].). We manually identified four species with high certainty: $N. noctula$ ($F_{end}$ <22 kHz), $P. nathusii$ ($F_{end}$ > 30 kHz and ≤42 kHz), $P. pipistrellus$ ($F_{end}$ >42 kHz and ≤51 kHz), and $P. pygmaeus$ ($F_{end}$ >51 kHz). Echolocation sequences with end frequencies >22 kHz and ≤29 kHz were classified as the Nyctaloid species group, which likely included $N. noctula$, $N. leisleri$, $E. serotinus$, $E. nilssonii$, and $V. murinus$, all of which are known to be susceptible to wind turbines[18,40]. Feeding buzzes were classified as belonging to the Pipistrelloid group when measured search flight calls did not allow unambiguous identification of one $Pipistrellus$ species. This was the case when one call sequence contained bat calls with frequency parameters that fit two $Pipistrellus$ species ($P. nathusii$ and $P. pipistrellus$ for measured end frequencies below and above 42 kHz or $P. pipistrellus$ and $P. pygmaeus$ for measured end frequencies below and above 51 kHz).

## Acoustic analysis of social vocalizations

Social vocalizations are often embedded in echolocation call sequences, thus facilitating species identification, and some social vocalizations themselves allow for correct species identification (see Supplementary Table 10 for details on literature used for species identification). We manually identified 4129 social vocalizations in 983 files. On average, these 983 files contained 4.2 ± 6.0 social vocalizations, ranging from 1 to 57 social vocalizations per file. We identified six species and one genus with high certainty: $N. noctula$, $N. leisleri$, $V. murinus$, $P. nathusii$, $P. pipistrellus$, $P. pygmaeus$, and $Plecotus$ spp. ($P. austriacus$ or $P. auritus$). When measured search flight calls did not allow unambiguous species identification, we classified social vocalizations as belonging to the Nyctaloid or Pipistrelloid group (comparable to the procedure for feeding buzzes).

Social vocalizations can be classified as calls or songs. Depending on the species, songs may contain a mix of single syllable types and stereotypic song elements (i.e., more than one syllable type combined into a stereotypic sequence) or only song elements that are produced in a repetitive manner and are sometimes interspaced with echolocation calls. $Plecotus$ spp. seems to be an exception to this rule as males only utter a single syllable type in rapid succession during their song flight[76] (Supplementary Table 2). We only classified a vocalization as song if it contained song elements (or, in the case of $Plecotus$ spp., characteristic repetitive syllables) and if it had previously been described to be associated with courtship and territorial behaviors (Supplementary Table 10). Moreover, for the Pipistrelloid group we only considered a vocalization to be song if at least five consecutive song elements were produced. In this group, a high repetition rate of song elements is indicative of song flight of courting males[27,31] while single song elements may also be produced by both sexes in the context of food patch defense[77,78]. To be conservative, we treated vocalizations from $Plecotus$ spp. in the same way as Pipistrelloid vocalizations because there was comparatively less information available on their song flight[76]. For all species, we considered song elements produced in successive sequences as 'song events' and determined their duration.

## Statistical analysis

We used generalized linear mixed models (GLMM) with a binomial error structure and a logit link function to investigate the effects of the species group (Nyctaloid or Pipistrelloid) and month on feeding and social activity (i.e., the proportion of social calls or feeding buzzes among all bat recordings). The model was implemented in R[79] using the glmer function from the lme4[80] package. Fixed effects included the main effects of species group and

month, as well as their interaction, to assess whether the effect of month differed between species groups. Month was modeled as an ordered factor with polynomial contrasts (up to the fifth degree) to capture potential non-linear trends. Initially, random intercepts for year, site, and an observation-level factor (to account for overdispersion) were included in the model. Year did not explain any variability and was subsequently removed from the model. To optimize model convergence, the bobyqa optimizer was used, and the maximum number of iterations was increased to 100,000. We excluded wind turbine-year-month observations with zero bat recordings from the binomial GLMM (11 excluded observations, 267 included observations), however, their inclusion did not affect model estimates. We assessed model fit using DHARMa[81] residual diagnostics, which indicated a good model fit, no dispersion issues, and no significant outliers for both GLMMs on feeding and social activity. We calculated marginal means for the interaction between species group and month using the emmeans[82] package, and Tukey-adjusted pairwise comparisons were used to test specific contrasts.

## Covered flight distance and acoustic signaling range of singing bats

We drew information on commuting flight speeds of the seven singing bats in our study from the literature (see Supplementary Table 6 for details). Combined with the duration of song events, commuting flight speeds enabled us to calculate the flight distance a singing bat covered during each song event. We subsequently tested if this distance was larger than the average detection range for species specific bat song (peak frequencies ranging from 14.0 kHz to 26.4 kHz) and call intensities (ranging from 72–108 dB peSPL at 1 m) of the acoustic recorder installed in the nacelle (at 20 °C and 60% humidity with a trigger threshold level of 37 dB SPL, see also Supplementary Tables 7 and 8 for information on bat song peak frequencies, call intensities and the estimation of the recorder's detection range). When the distance covered during a single song event was larger than the detection range of the recorder for species-specific bat song, we inferred that singing bats were not simply passing the turbine but prolonging their stay in the rotor swept area, for instance by circling the turbine tower or the nacelle.

Due to the attractive function of bat song[28,44], we wanted to know the distance over which males singing at wind turbines may be heard by conspecifics. We calculated the acoustic signaling range of bat songs with a formula originally developed for calculating the maximum detection distance of objects by echolocating bats[83]. Species-specific call intensities were obtained from the literature (see Supplementary Table 9). We ran calculations for an open space with a humidity of 60% and a temperature of 20 °C, which reflected typical ambient conditions around the nacelle[34,84]. As for echolocating bats, we assumed a detection threshold of 20 dB SPL but, contrary to echolocating bats, we doubled the calculated detection distance because song echoes do not need to travel back to the emitter but are picked up by a receiver instead. This approach has been used for calculating the active space of bat songs in the past[38].

## 3D thermal imaging of bat density in the rotor swept area

Using 2D thermal imagery, a study from North America provided evidence that bats actively approached wind turbines performing close approaches, flight loops and dives, as well as hovering and bat chasing[26]. If bats are attracted to and interact with wind turbines, we predicted that bat density should increase near the nacelle and within the rotor swept area of a turbine. To estimate the density distribution of bats as a function of their distance to the turbine nacelle we analyzed stereo-thermal recordings of bat flight paths from six nights at four wind turbines from two sites and a total of 29.9 h in 2008 and 2012 (Supplementary Table 11). The rotor radius of studied wind turbines was 35 m. We used Quantum Well Infrared Photodetector cameras (AIM 640Q, AIM Infrarot-Module GmbH) which consisted of 640 × 480 infrared detector elements (approximately 0.3 megapixel) and used a 50 mm lens. This resulted in a field of view of 17.5° × 13.1°. The temperature resolution was >30 mK, which means that an object differing by 0.03 K from background temperature should be detectable in the recorded images. Images were recorded with 14 bit

and 11 frames/s. The Fraunhofer Institute of Optronics, System Technologies and Image Exploitation (IOSB) calculated the range performance using the heat of an object, the resolution, and the thermal sensitivity of the camera. During good weather conditions, a bat (i.e., represented by a sphere of 5 cm diameter and a 4 K temperature difference to the background) could be detected at 500 m distance to the camera.

Stereo recordings are necessary for determining the 3D positions of objects. For stereo images, two cameras are positioned at a defined distance so that their fields of view overlap (individual fields of view of cameras are depicted in light pink and light blue in Supplementary Fig. 14). In the area where the fields of view overlap, the distance of an object from the camera can be calculated. We positioned the two cameras approximately 200 m from the turbines with a base distance of 16 m to each other (see Supplementary Fig. 14A for an illustration of the camera set-up) resulting in a distance resolution of 1.5 m. The thermal cameras were positioned to focus on the identical point in the optical center of each camera (orange dot in Supplementary Fig. 14) and horizontally leveled using a bubble level. The two-dimensional detection of bat flight trajectories in the recorded sequences from each of the thermal cameras was achieved by manually controlling difference images using LabVIEW (National Instruments, Austin, TX). The difference formation of successive images suppresses stationary objects, such as the mast and the nacelle of the wind turbines and highlights moving objects, such as the rotor and flying animals. Both thermal cameras were linked to a single frame grabber, which was operated via LabVIEW to ensure synchronized frame acquisition.

Using the known geometry of the stereo thermal camera array, the three-dimensional movement of an animal was calculated from the two 2D flight trajectories. The triangulation required the base distance, the angles of the cameras, the focal length of each camera and the position of the bat in each image (Supplementary Fig. 14B). The calculation of the three-dimensional position was done using trigonometric functions. Vertical height and horizontal level are identical and constant during the conducted measurement periods. The triangulation of the matching detections of the left and right camera was implemented in Matlab (version 2012b, Mathworks, Inc., Natick, MA, USA).

Using angular triangulation without considering lens distortion and intrinsic and extrinsic camera matrices is prone to error. To minimize these systematic errors, 3D positions of bats were related to a triangulated point (orange dot in Supplementary Fig. 14) at the nacelle of the wind turbine. Using this normalization minimized systematic errors that occurred during the measurement of the relative locations of both cameras. This was verified using a GPS equipped drone (OktoXL by MikroKopter). The drone did not require additional heating, as the heat generated by the rotors and electronics was sufficient for localization with the thermal cameras. Localization with GPS is also prone to error especially at the drone´s flight height. Nevertheless, using the drone's GPS as ground truth allowed us to investigate potential systematic errors of the stereo-camera system. Supplementary Fig. 15A shows the flight trajectory of one test flight of the drone around a wind turbine. We evaluated the 2D distance (top-down projection, without height coordinate, Supplementary Fig. 15B) as well as the 3D distance between drone and nacelle (Supplementary Fig. 15C). Comparison between several GPS and thermal stereo localizations showed minor differences with no tendency towards strong systematic errors around the turbine.

Flight trajectories recorded with the 3D thermal imaging system were correlated with acoustic detections at nacelle height. The majority of recorded flight trajectories approaching the nacelle to less than 20 m were confirmed as bats by acoustic recordings (76%). At distances larger than 20 m to the acoustic detectors the probability of recording bat calls, especially higher frequency echolocation calls, decreases significantly[85] (Supplementary Table 8). Therefore, and to avoid biasing our data analysis for flight trajectories recorded closer to wind turbines we assumed that all recorded flight trajectories were from flying bats when calculating bat density around the nacelle (see Supplementary Table 12 for details). Supplementary Figs. 16 and 17 show two examples of reconstructed flight trajectories of bats flying within the rotor swept zone.

The stereo system was then used to determine bat density around the nacelle of wind turbines. Bat localizations were referenced to the principal point (orange dot in Supplementary Fig. 14). Bat positions were counted according to their distance to the principal point. Proceeding from the center of the nacelle, bat positions were grouped in spherical shells with a thickness of 3 m. (Fig. 5). The three-dimensional volume of the shells was calculated using geometric principles considering the cut-off of certain shells due to the viewing angle of the thermal cameras. We then calculated bat density by distance to the nacelle as the number of bat positions divided by spherical shell volume.

We verified the robustness of our bat density calculation through the separate analysis of different spatial areas (see Supplementary Fig. 18). Although the cameras did not cover the entire rotor-swept area, comparable exponential decreases in bat density with increasing distance from the nacelle were observed in front as well as behind the wind turbine. The lower bat density behind the wind turbine can be explained by a general reduction in detection probability with greater distance from the cameras and by lower activity of all bat species at higher altitudes[57,86]. The area behind the wind turbine covered by the cameras was higher above ground than the area in front of the turbine.

## Statistics and reproducibility

We analyzed feeding and social activity of bats with generalized linear mixed models (GLMMs) with a binomial error structure and logit link function using the lme4 package in R. Model fit was assessed with DHARMa residual diagnostics, and marginal means with Tukey-adjusted pairwise comparisons were obtained using the emmeans package.

Acoustic data were recorded for 26 turbine–years (22 turbines sampled across 6 sites and five natural regions). 83,292 of 107,950 acoustic recordings were manually verified to contain bat calls. We identified feeding buzzes ($n = 1598$) in 1196 call files, and social vocalizations ($n = 4129$) in 983 files. Replicates consisted of independent acoustic recordings collected across nights ($n = 4431$), turbines ($n = 22$), and years ($n = 3$), with turbine-year ($n = 26$) treated as the primary unit of replication. We gathered thermal 3D flight trajectories of bats from six nights at four turbines (29.9 h total). Reproducibility was supported by standardized long term sampling across the activity period of bats, independent replication across sites and years, validation of species identification, and 3D localization accuracy (Supplementary Figs. 14–18). Measurements not subject to formal statistical analysis were evaluated descriptively.

## Ethics statement

This study involved non-invasive acoustic monitoring and thermal recordings of free-flying wild bats. No animals were captured, handled or experimentally manipulated. Therefore, the study did not require approval from an animal ethics committee under German regulations. We have complied with all relevant ethical regulations for animal use.

## Reporting summary

Further information on research design is available in the Nature Portfolio Reporting Summary linked to this article.

## Data availability

The numerical source data underlying all graphs and charts presented in this study are available in an open-access repository via Figshare at https://doi.org/10.6084/m9.figshare.31112509[87]. Additional data supporting the findings of this study are available in the Supplementary Information or from the corresponding author upon reasonable request.

## Code availability

No custom code was generated for this study. All analyses were performed using standard, publicly available software packages, including R (versions 4.3.1 and 4.4.3), as well as Avisoft RECORDER (version 4.2, Avisoft Bioacoustics, Glienike, Germany), Avisoft SASLab Pro (versions 5.2.13 and 5.2.14, Avisoft Bioacoustics, Glienike, Germany), LabVIEW (National

Instruments, Austin, TX), and Matlab (version 2012b, Mathworks, Inc., Natick, MA, USA).

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

## Acknowledgements

We thank the Project Management Jülich particularly F. Klein for project management. We express our gratitude to the operators of wind turbines for granting us access to their turbines. We thank our project partner ENERCON GmbH (especially B. de Wolf, K. Einnolf, F. Kentler, U. Kleinoeder, M. Schellschmidt, R. Schulte, and several service teams) for technical support and for the installation of acoustic detectors and sensors at the wind turbines. We are grateful to K. Jung for providing the custom-made identification key of bat echolocation calls. We thank the German Federal Ministry for the Environment, Nature Conservation and Nuclear Safety (BMU, funding code 0327638C+D) and the German Federal Ministry for Economic Affairs and Energy (BMWi, funding code 0327638E) for funding awarded to O.B. We would also like to thank three anonymous reviewers for their most helpful comments on the manuscript.

## Author contributions

M.K. and M.N. designed the study. K.H., M.N., N.W., O.B., and R.S. collected the data. C.H., K.H., M.K. and M.N. analyzed the data. M.K. and M.N. wrote the manuscript. All authors contributed to the final version of the manuscript.

## Funding

## Competing interests

The authors declare no competing interests.
