## [Transparent Peer Review file · Communications Biology]

Song flight and 3D thermal detection provide evidence for bat attraction to wind turbines in Central Europe

Corresponding Author: Dr Martina Nagy

Version 0:

Reviewer comments:

Reviewer #1

(Remarks to the Author)

Dear associate Editor, dear authors,

I thank you for asking me to review this article, which I found interesting and enjoyed reading. The paper presents results from the analysis of passive acoustic monitoring data of bat activity in wind turbines, collected at the nacelles' height. These are presented together with a dataset of bat position data obtained with thermal cameras. The authors claim that a set of bat species (and groups of species), detected at the space within reach of the rotor of wind turbines, are attracted to these structures because they make use of them as sites for social and mating behaviors. This claim is based mainly in the detection of song flight, the fact that the average duration of songs is longer than what would be detected if individuals were flying across instead of sticking around in the vicinity and within the rotor-swept area, and an increase in the density of bat positions recorded with thermal cameras with decreasing distance to the wind turbine's nacelle. Although the results are descriptive, they provide very useful evidence for species-specific bat attraction to wind turbines, in a region that is experiencing a strong increase in the implantation of these structures.

I believe the authors' intent is also to inform possible impact mitigation strategies, such as algorithm-based curtailment. Conditions for feathering the rotors could for instance be set to vary depending on whether it is the mating season for the affected species. This point could've been more highlighted in the Discussion. The authors will notice that throughout the manuscript, many of my detailed comments regard the redundancies, abundance of figures and technicalities that can make the article a bit heavy to read, hindering a message that otherwise could be quite straightforward.

It's striking that a large portion of the cited references are gray literature, and often in German. Although bat activity at wind turbine nacelle height remains poorly studied, I've added a few reference suggestions in my detailed comments that could perhaps enrich the bibliography, please feel free to add or ignore them.

Detailed comments and suggestions by line:

L1 the title and abstract mention 3D flight paths, but it's a bit disappointing not to find any illustration or analysis of actual trajectories. Instead, if I understood well, the density of recorded bat positions was analyzed and illustrated in figure 6. I suggest that the authors rephrase the title to rather highlight the combination of acoustic and thermal detection techniques, something in the line of "Acoustic and thermal bat detection reveal behaviors associated to bat attraction to wind turbines".

L21 "...but its negative..."

L22 "Fatal interactions" may be substituted by "Collisions" throughout the manuscript to keep a coherence in terms.

L33 see also <https://www.sciencedirect.com/science/article/pii/S0048969723000190>

L40 May remove "Non mutually exclusive"

L64 to 84 The introduction is a bit lengthy. These two paragraphs could be merged and shortened.

L86, 87 This information can be moved to the Methods section.

L99 to 116 I think this belongs in the Methods section. Please remove repeated information between the Methods and Results section.

L121 to 124 This information is not particularly interesting in my opinion. Removing these details from the results may help the reader focus on the important parts.

L135 remove "might"

L158 to L163 perhaps it's better to put this in the Discussion section instead of the Results section.
L165 to L169 this is redundant with L436 to 446 in the Methods section, and can be removed.
L186, 187 Also redundant with the Methods section.
L194 to L200 I think this belongs in the Methods section, and may be inserted before L456.
L204 Do you mean Fig.6?
L206,207 I think this belongs in the Discussion section.
L216 to 220 These sentences may be removed, they repeat too much the results.
L240 "song elements became louder and fainter [...]" Can the authors think of a way to illustrate this in a more conspicuous way than just by observing figure S4? I'm just not sure it's enough to be able to say this in the discussion.
L242 "The presumed song flight trajectories". I don't think this is well illustrated by the data visualizations of the manuscript. I am missing an illustration or an analysis of actual reconstructed bat trajectories. I'm not demanding they do it, but if the authors choose not to, references to trajectories should be removed from the manuscript altogether, probably also the proposed trajectories in figure 5D.
L259 "[...] poor environment adds apparently suitable [...]"
L266, 267 see also <https://onlinelibrary.wiley.com/doi/10.1111/1365-2664.14227>
L290 see also <https://www.sciencedirect.com/science/article/pii/S000632071730229X>
L315 "[...] might help explain [...]"
L335 It's a bit strange to finish the discussion like this. The authors invoke possible "learning" mechanisms that change the attraction of bats to wind turbines over time, but provide no published evidence of such mechanisms even being possible. Moreover, they omit the inverse scenario, which may well be more likely, where a sustained attraction to wind turbines would imply these act as ecological traps. Such traps may only be mitigated through the implementation of well-parameterized curtailment. I am missing a bit more development of these points.
L376 & 379 see comment on L290
L404 reference 14 is relevant here
L447 to 454 (applies to corresponding results and discussion) I'm not convinced that this information is interesting. If the authors choose to keep it, they should justify its relevance here from a biological or conservation point of view. Personally, I would remove it to make the article more concise. The point of song behavior attracting further bats to rotor-swept zones can still be made in the discussion.
L462 "leading to a data rate of about 23 GB/hour" can be removed, as it is not necessary for replication.
L502 replace "not bias" by "avoid biasing"
L508 "spherical shells"
L509 I'm not sure I understand well, what do the authors mean by "straight forward"?
L511 "We then calculated bat density by distance to the nacelle as the number of bat positions divided by spherical shell volume".
Figure 1 the main file contains quite a lot of figures, perhaps this first one can go to the supplementary material, since it is not really important for the Discussion section.
Figures 2, 4 Reference to the natural regions of Germany may be removed to make the figures less crowded.
Figure 3 "Pipistrelloid species showed proportionally higher social activity [...]" The authors could actually run a statistical test for this (e.g. a beta regression), just to be more confident in the significance of the difference for August.
Figure 4 The lower panel containing song excerpts can be moved to the supplementary material, and the upper panel can be combined with figure 2, reducing the total number of figures in the article by yet one less.
Figure 5 N leisleri, Pipistrellus sp. and Plecotus sp. have very few observations. They could be left out, making the figure less crowded, and the results easier to interpret. In the caption, line 789, do the authors mean Table S5?
Figure 6 (panel B) the circle corresponding to the limit of the rotor-swept zone could perhaps be highlighted in another color. Also, in panel C, the limit of the collision-risk zone could be indicated with a vertical red line at the 35m distance to the nacelle position, for example.
Figure S2 Instead of enumerating the vocalization types in the caption, they could be labelled directly on the figure.
Figure S5 Could the authors keep the same axis dimensions (not the scale) for panel E as the rest of panels?
Figures S6, S7, S8 What is the scale of the Y axis?
Figure S10 this figure is redundant with Figure 3. The authors could either show October in figure 3, or just remove this figure, since, as is mentioned in the caption, information on this month is not very interesting due to uncertainty in the mean.
Figure S12 I'm not sure that the German landscape classification adds much information to a non-German reader. The map could be simplified removing these subdivisions.
Figure S13 Could the authors please describe the figure panels (A and B)?
Figure S14 Same comment as for figure S13
Table S2 This table might be more compact and easier to read if the species are placed as columns, avoiding the repetition of syllable/song element types.
Table S4 As in figure 5, data for N leisleri and Plecotus sp are insufficient to be considered representative, and can be left out.
Table S5 A and B are quite separate tables, they may be numbered separately.
Table S9 caption, lines 195 – 197 ("Therefore [...] nacelle"). This information is already given in the Methods section, so this part of the caption may be removed.

Reviewer #2

(Remarks to the Author)

This manuscript came as a shock to me. Having been deeply involved in the initial response and scientific investigation into

bat fatalities at wind turbines in North America, about a decade ago I retreated from active research into the underlying causes of this novel and deeply complex problem.

I withdrew in response to growing obstacles to testing attraction hypotheses and frustration that North American researchers and their ubiquitous industry and/or government regulatory partners seemed more focused on engineering solutions to the bat-turbine problem than investigating its root causes and better understanding the behaviors and natural histories of the bats in question.

Research into machine-based engineering solutions and turbine operational changes has been consistently prioritized in the U.S. and Canada over basic biological research and what I consider a life-history perspective. As a result, we still lack clear and evidence-based scientific explanations for why the proportionally few species of bats most affected by wind turbines in the U.S. and Canada (some of which may go extinct because of wind turbines) seem attracted to the very structures we are working so hard to keep them away from with operational changes and new mechanical inventions. That could all change with the publication of this study.

The elephant in the room, which the authors diplomatically refrain from bringing up in this exemplary manuscript, is that those of us in the scientific weeds of the problem in other parts of the world lag far, far behind in having the means to test, let alone carry out, meaningful observational studies of bats around operational wind turbines. The authors seem to have taken the opportunities and information available to them and done everything right. This paper will undoubtedly be a surprising game changer with global impact.

I've never used such a glowing adjective in my 30+ years of reviewing scientific papers, but I view this manuscript and study as a masterpiece. Without hyperbole, it reports findings based on clear and sufficient evidence of a nature that I never expected to see during my scientific career. I knew that the authors had long been deploying good bat detectors on wind turbines across Germany; I knew that there was a rich background of natural history observations and understanding of European bat behaviors that went far beyond the usual focus on those occurring in nice weather, close to the ground, and in beautiful natural areas; and I knew that there was more focus on investigating the underlying causes of bat fatalities at wind turbines in Europe than other regions. I did not know the methods and technologies for recording, analyzing, calibrating, bias correcting, validating, and interpreting these rare observational events had been so carefully and meticulously developed, tested, and implemented. I found no holes in the assumptions, reasoning, data analyses, validation methods, or inferences drawn despite reading the manuscript carefully twice and skimming it a couple times more.

The writing is clear and concise, the figures are beautiful, and the legends provide sufficient information for the reader to easily grasp their meaning at a glance while drawing deeper understanding with more careful reading. The various analytical frameworks used for interpreting the 2D and 3D spatial distributions of sound and visual signals are creative and more easily repeatable by future investigators than any I have yet seen. In general, all the methods seem easily repeatable for future studies given the level of detail provided. The illustrations of feeding buzzes, social call varieties, and songs among several species will undoubtedly serve to educate and surprise many readers and scientists, including bat researchers, who were ignorant of their existence. I think the authors essentially 'write the book' about how this kind of study can be done and then reported very successfully. Even the supporting information impressed me. For example, Table S9 showing the inferred differences between optical flight observations and acoustic detections of bats around turbines is something I have personally attempted and completely failed at several times, despite great cost and effort. So subtle a table--such valuable information.

Aside from a few rare editorial glitches (see detailed comments that follow), the only substantive suggestion I have for improving the text is to add a bit of detail about how the thermal cameras were temporally synchronized for the 3D analysis. I'm guessing they were tethered to the same capture device or that their clocks were somehow regularly synchronized? I also wondered if the drone was supplementally heated, or if its regular operational waste heat was sufficient to detect with the thermal cameras...very minor questions...and few.

Another suggestion I am still on the fence about making is whether to recommend going a bit further out into speculation in the discussion section. The lack of physical evidence for song flight in the North American species of bats most affected by wind turbines (*Lasiurus cinereus*, *L. borealis*, *Lasionycteris noctivagans*) isn't likely due to it not occurring, but because few people have ever looked for it and, if found, characterized social calls and songs in these species of bats. Three studies that stand out in this regard that were not cited hint at the possibility that something similarly complex might be happening in North America. However, we lack the background natural history observations and have thus far missed ample research opportunities to uncover such connections. These papers were:

Reyes, G.A., Szewczak, J.M. Attraction to conspecific social-calls in a migratory, solitary, foliage-roosting bat (*Lasiurus cinereus*). *Sci Rep* 12, 9519 (2022). <https://doi.org/10.1038/s41598-022-13645-9>

Corcoran, A.J., Weller, T.J., Hopkins, A. et al. Silence and reduced echolocation during flight are associated with social behaviors in male hoary bats (*Lasiurus cinereus*). *Sci Rep* 11, 18637 (2021). <https://doi.org/10.1038/s41598-021-97628-2>

Cryan P.M., Jameson, J.W., Baerwald, E.F., Willis, C.K.R., Barclay, R.M.R., Snider, E.A., et al. Evidence of late-summer mating readiness and early sexual maturation in migratory tree-roosting bats found dead at wind turbines. *PLoS ONE* 7(10): e47586 (2012). <https://doi.org/10.1371/journal.pone.0047586>

These scrappy observations from elsewhere may or may not be worth bringing up in the discussion, since any well-informed readers that see this paper in North America will grasp the relevance and possible connections. However, those of us in

North American would welcome some advice about how to go about looking into these questions by those who succeeded in doing so.

Another thing I wondered about while pouring over the figures and tables about the seasonal prevalence of song flight and social behaviors in the Pipistrelloid and Nycaloid groups was whether the authors think the patchiness of social behaviors observed among sites and thus regions might have to do with migratory tendencies of certain group members? For example, does the higher prevalence of Pipistrelloid social calls and songs at turbines in August through October reflect some species of Nyctaloids being more likely to leave Germany in late summer for mating and wintering areas elsewhere? It seems that being in the right place at the right time, or vice versa, might further help explain those broad seasonal pattern differences among feeding and social detections? As my poorly articulated question indicates, this may not be an easy question to address without a lot of text, but I'm bringing it up here in case there might be a concise way of touching upon it.

My minor editorial notes:

Line 177: is "spec" the journal standard for species or is it spp?

Line 209: is the plural of nacelle nacelles?

Line 245: "...or the edge[s] of woodlands..."

Lines 256-257: "...song flight is performed in [close proximity to] mating roosts and flight paths also often follow [linear or prominent landscape features] such as [forest edges] or buildings..."

Line 315: "...might help [explain] the highly female-biased fatality ratios."

Line 360: is this time in UTC?

Line 371: maybe add a sentence about what the 'BMU-settings' are for readers without access to the software or documentation?

Line 421: comma after i.e.?

Line 445: "...species-specific bat song, [we inferred that] singing bats were not simply passing..."

Line 511: hyphenate? distance-dependent bat density

Page 14, Line 123: "Wind turbine sites were sample[d] in four different..."

Page 14, Lines 125-126: hyphenate so-called?

Page 16, Lines 137-138: "No systematic errors were detected[. Mismatches illustrated by non-overlapping sections of the blue and gold lines resulted] from the accuracy of the GPS [and] two-dimensional detection [error in determining] the drone's center."

Reviewer #3

(Remarks to the Author)

The manuscript is well written adds to the growing body of knowledge regarding bat interactions with wind turbines. Below are a few minor suggestions and a few moderate to major suggestions to improve the manuscript.

Line 28: There are more recent syntheses of curtailment studies available. See Adams et al. 2021 'A review of the effectiveness of operational curtailment for reducing bat fatalities at terrestrial wind farms in North America' and Whitby et al. 2021 'The state of the science on operational minimization to reduce bat fatality at wind energy facilities.

Line 32-33: The reports on the 'cost' of curtailment are in terms of annual energy production and not revenue. See Whitby et al. 2022 for more information on the loss of annual energy production from curtailment in the US.

Line 41 and line 52: Suggest adding Guest et al. 2022. 'An updated review of hypotheses regarding bat attraction to wind turbines'.

Line 54: There is a more recent analysis on the relationship between pre-construction activity and post-construction fatality. See Solick et al. 2020 'Bat activity rates do not predict bat fatality rates at wind energy facilities'.

Relating to section 'Bat songs at wind turbines' starting on line 137: 1) How many nights were social calls recorded? How many social calls/night? On nights with social calls, was there an increase in bat activity relative to non-social call nights (this would help make the connection that social calls are attracting bats)? How does the rate of social call activity at wind turbines relate to airspace without wind turbines (does it occur more often or less often)? I realize your study may not be able to answer the last one because of how the study was set up, but if there is no difference in social call activity from wind turbines to other areas then it cant be considered an attractant.

Line 235: 'considerable distances'. As stated above, 42 m and 100 m are not considerable distances. To hear these calls, bats would already have to be within close proximity to the turbine.

Line 247: 'frequently'. Are these calls frequent? They don't appear to be based on the data presented.

Line 266: '..are as yet the strongest evidence for bat attraction'. I disagree that it is the strongest. It is additional data to a growing body of knowledge that points to certain species of bats being attracted to wind turbines/spending more time around wind turbines.

Line 190: distance of social calls = 42 m for *Plecotus* spp. and 100 m for *P. nathusii*/ *V. Murinus*. These are not long distances. For *Plecotus*, another bat hearing the call would already have to be near the wind turbine already. The turbines have an approx 35 m blade length, so if a male *Plecotus* is singing at the nacelle, a listening bat would be only 7 m away from the tip of the blade. Similarly, for *P. nathusii*/*V. Murinus*, though the distance is longer, it's still relatively short. This really eliminates any idea that bats are being drawn into the area by hearing calls. They are already close by. Now, it may increase the flight times of bats in and around wind turbines, thus increasing their risk. But it's not a viable cause of attraction.

Line 214-215: 'Bat song was documented at all studied sites and in 88% of the 26 wind turbine-years, suggesting that it is not a rare behavior, but that bats commonly sing at wind turbines in Germany.' It certainly seems like a rare behavior. What is the percentage of calls of singing. If I understand it correctly out of approximately 83,000 recordings, 4,100 were social vocalizations, and 1,300 were songs. That doesn't seem like a common occurrence. For example, say the night is 8 hours long, and a song is 1 minute and there is only 1 song per night, that leaves 539 minutes of time when songs are not occurring/being heard. So, other bats would have to be within hearing distance at the exact 1 minute interval across the entire night to be attracted to it.

Lines 278-281: '...it appears that evidence in favor of the feeding attraction hypothesis is increasing'. This makes it sound like the feeding hypothesis is out-competing other hypotheses. Rather, it just points to the conditions that occur across your sample turbines. At one point it is mentioned that the turbines are on open crop land. That would be primary foraging habitat for bats regardless of the presence of wind turbines. So, to say that bats are foraging more around wind turbines in this landscape relative to other airspace, you'd need to have detectors at turbine and non-turbine locations. It would also be helpful to determine whether there are more insects in and around turbines relative to open airspace. Moreover, the surrounding habitat likely plays a role in how bats perceive wind turbines. An agricultural setting with new tall structures could present new roosting opportunities to bats as compared to a forested habitat. Alternatively, a forest habitat with new clearings for turbines and roads, may create new edge habitat for foraging and commuting. All this is to say that traction hypotheses are not mutually exclusive, they are likely occurring at different scales, differ by species, differ by habitat, and possibly interacting in ways that we still don't fully understand.

Figure 3: This speaks to how rare the feeding and social calls are. At no point do feeding buzzes exceed 5% even at the higher 95%CI. Similarly for social vocalizations (though in August the higher 95% gets slightly above 5%). Social vocalizations for *Nyctaloid* are near zero for 3 of the 5 months. August and September show the highest percentage of social vocalizations for *Pipistrelloid*. You would expect to see an increase in overall *Pipistrelloid* activity associated with these data if social vocalizations are attracting bats within this group.

Figure 6, cameras set up and sample size: It's difficult to make any definitive statements about the results from 6 nights of data collection across 4 wind turbines. That is a rather small sample. Moreover, the field of view is heavily skewed toward detecting bats close to the wind turbine because the overlap of the 2 cameras is centered on the nacelle. If the field of view covered the entire 60 m concentric circles, that would be one thing, but Fig. 6 shows that most of the farther distances are not even in the field of view. The cameras don't even cover the entire rotor-swept area. It's also not surprising that you are seeing fewer observations the farther away from the camera as detection decreases with distance.

Version 1:

Reviewer comments:

Reviewer #1

(Remarks to the Author)
Dear Authors, Dear Editor,

Thank you for your patience in waiting for my report. I find the manuscript has gained in clarity concerning its main claims and the supporting evidence. I'd also like to commend the authors for their efforts to provide all the necessary details for study reproducibility, which is something we often lack.

I personally would've preferred to see a results section that avoids giving methods information (e.g. lines 143-151 and 173-179) and rather synthetically presents the evidence. Also, the paragraph in lines 260-268 is not very well woven into the discussion. Still, these points are not really important, and it is above all desirable to see this article published soon so that the community can finally have access to it. I find no major issues to be reported, and am happy to recommend its publication in *Communications Biology*.

Reviewer #3

(Remarks to the Author)

The authors did a great job of addressing my comments. I really appreciated their thoughtful responses and how their efforts to revise the manuscript. I have no further comments. Great job.

Reviewers' comments:

Reviewer #1 (Remarks to the Author):

Dear associate Editor, dear authors,

I thank you for asking me to review this article, which I found interesting and enjoyed reading. The paper presents results from the analysis of passive acoustic monitoring data of bat activity in wind turbines, collected at the nacelles' height. These are presented together with a dataset of bat position data obtained with thermal cameras. The authors claim that a set of bat species (and groups of species), detected at the space within reach of the rotor of wind turbines, are attracted to these structures because they make use of them as sites for social and mating behaviors. This claim is based mainly in the detection of song flight, the fact that the average duration of songs is longer than what would be detected if individuals were flying across instead of sticking around in the vicinity and within the rotor-swept area, and an increase in the density of bat positions recorded with thermal cameras with decreasing distance to the wind turbine's nacelle. Although the results are descriptive, they provide very useful evidence for species-specific bat attraction to wind turbines, in a region that is experiencing a strong increase in the implantation of these structures.

I believe the authors' intent is also to inform possible impact mitigation strategies, such as algorithm-based curtailment. Conditions for feathering the rotors could for instance be set to vary depending on whether it is the mating season for the affected species. This point could've been more highlighted in the Discussion. The authors will notice that throughout the manuscript, many of my detailed comments regard the redundancies, abundance of figures and technicalities that can make the article a bit heavy to read, hindering a message that otherwise could be quite straightforward.

It's striking that a large portion of the cited references are gray literature, and often in German. Although bat activity at wind turbine nacelle height remains poorly studied, I've added a few reference suggestions in my detailed comments that could perhaps enrich the bibliography, please feel free to add or ignore them.

Dear Reviewer 1:

We would like to express our sincere gratitude for the thorough review of our manuscript and for your thoughtful and constructive feedback. We have adopted most of the suggestions regarding the removal of redundancies and the reduction of figures in the main manuscript, and we have incorporated all suggested citations.

In addition, as recommended, we have conducted GLMMs on the feeding and social activity data. These analyses now provide statistical support for our finding that Pipistrelloid

species exhibit more social activity at wind turbines than Nyctaloid species, particularly in late summer and autumn.

We have also revised the Discussion to integrate the concept of wind turbines potentially acting as ecological traps for bats and to further emphasize the critical role of algorithm-based curtailment in mitigating negative impacts on bat populations.

Once again, we sincerely appreciate your valuable insights, which have helped us improve the clarity and impact of our manuscript.

Detailed comments and suggestions by line:

L1 the title and abstract mention 3D flight paths, but it's a bit disappointing not to find any illustration or analysis of actual trajectories. Instead, if I understood well, the density of recorded bat positions was analyzed and illustrated in figure 6. I suggest that the authors rephrase the title to rather highlight the combination of acoustic and thermal detection techniques, something in the line of "Acoustic and thermal bat detection reveal behaviors associated to bat attraction to wind turbines".

Line 1: We have rephrased the title: *Song flight and 3D thermal detection of bats provide evidence for bat attraction to wind turbines in Central Europe*

In addition, we have added two examples of reconstructed flight trajectories to the supporting information (figures S16 and S17), one example of a suspected inspection flight and one example of a probable collision.

L21 "...but its negative..."

Line 21: Done

L22 "Fatal interactions" may be substituted by "Collisions" throughout the manuscript to keep a coherence in terms.

Lines 23-25: We have added a sentence on lines 23ff, to explain that we are using these terms inter-changeably: "based on the assumption that bats colliding with the rotor are unlikely to survive we use the terms collision, mortality, and fatality inter-changeably

L33 see also <https://www.sciencedirect.com/science/article/pii/S0048969723000190>

Line 35: We now also refer to the suggested citation.

L40 May remove "Non mutually exclusive"

Line 44: Done

L64 to 84 The introduction is a bit lengthy. These two paragraphs could be merged and shortened.

Thank you for your suggestion. However, we believe that both paragraphs serve distinct and important purposes — one outlining the existing hypotheses and the other detailing bat mating behavior. This context is crucial for understanding the rationale behind our study, so we prefer to retain them in their current form.

L86, 87 This information can be moved to the Methods section.

Line 96: We have removed the detailed second part of the sentence.

L99 to 116 I think this belongs in the Methods section. Please remove repeated information between the Methods and Results section.

Line 110, and line118: We have removed the information that was redundant between the Methods and Results sections.

L121 to 124 This information is not particularly interesting in my opinion. Removing these details from the results may help the reader focus on the important parts.

Lines 123: We removed the sentence from the manuscript.

L135 remove “might”

Lines 130-141: This paragraph has been rephrased and thus, the term “might” is not part of the paragraph anymore.

L158 to L163 perhaps it’s better to put this in the Discussion section instead of the Results section.

Lines 163-168: we would like to keep this short explanation for the result in the results section

L165 to L169 this is redundant with L436 to 446 in the Methods section, and can be removed.

Line 170: we have removed the redundant information.

L186, 187 Also redundant with the Methods section.

Line 186: We removed the redundant information.

L194 to L200 I think this belongs in the Methods section, and may be inserted before L456.

Lines 498-504: We moved the text to the methods section.

L204 Do you mean Fig.6?

Line 196: Yes, thank you. Please note, that we have moved former Figure 1 to the supporting information and this is Fig. 5 now.

L206,207 I think this belongs in the Discussion section.

Lines 198-200: We would like to keep this sentence here. Although it is redundant we believe it is helpful to have some interpretation of results already in the results section.

L216 to 220 These sentences may be removed, they repeat too much the results.

Lines 209-213: In the first paragraph of the discussion, we summarize our results on social vocalizations and specifically on songs of bats at wind turbines. Thus, we would like to stick to the redundancy.

L240 “song elements became louder and fainter [...]” Can the authors think of a way to illustrate this in a more conspicuous way than just by observing figure S4? I’m just not sure it’s enough to be able to say this in the discussion.

Figure S5: Thank you for this suggestion. In the revision, we calculate and plot peak-to-peak amplitudes for the song elements of the three depicted bat species. These curves illustrate that the loudness waxes and wanes, indicative of a bat moving around the microphone.

L242 “The presumed song flight trajectories”. I don’t think this is well illustrated by the data visualizations of the manuscript. I am missing an illustration or an analysis of actual reconstructed bat trajectories. I’m not demanding they do it, but if the authors choose not to, references to trajectories should be removed from the manuscript altogether, probably also the proposed trajectories in figure 5D.

Thank you for this important suggestion. We report the total number of reconstructed flight trajectories in table S11 and have now added two examples of reconstructed flight trajectories to the supplement (figs. S16 and S17) to illustrate this.

L259 “[...] poor environment adds apparently suitable [...]”

Line 253: Done

L266, 267 see also <https://onlinelibrary.wiley.com/doi/10.1111/1365-2664.14227>

Line 270: We added the suggested citation

L290 see also <https://www.sciencedirect.com/science/article/pii/S000632071730229X>

Line 294-295: We added “(e.g. feeding rates could be recorded at wind masts before turbine construction⁵⁷).” and the suggested citation.

L315 “[...] might help explain [...]”

Line 325: Done

L335 It’s a bit strange to finish the discussion like this. The authors invoke possible “learning” mechanisms that change the attraction of bats to wind turbines over time, but provide no published evidence of such mechanisms even being possible. Moreover, they omit the inverse scenario, which may well be more likely, where a sustained attraction to wind turbines would imply these act as ecological traps. Such traps may only be mitigated through the implementation of well-parameterized curtailment. I am missing a bit more development of these points.

Lines 334-340: Many thanks for these important suggestions. We have modified this paragraph to include the possibility of wind turbines acting as ecological traps for bats. We now write that:

“Attraction of bats to wind turbines in search of mating and foraging opportunities may lead to increased mortality, which, over time, may be reflected in decreased acoustic activity, turning wind turbine sites into ecological traps for these species (i.e., habitats that attract organisms but reduce their survival or reproduction; e.g.^{65, 66}). However, it is also possible that acoustic activity at turbines may initially increase during the first years following turbine construction if local and/or migrating bats need some time to discover those new sites. “

And we end the discussion and this paragraph with a suggestion to implement smart curtailment (lines 348-350): “In summary, we believe that only well parametrized curtailment (e.g.^{10, 11, 67}), incorporating stricter curtailment during the mating period, will effectively prevent bats from being attracted into an ecological trap.”

L376 & 379 see comment on L290

Lines 399 and 402: Thank you, we have added the citation and have added “at nacelle height” in both places, to clarify that we are talking about bat activity at height.

L404 reference 14 is relevant here

Line 424: Done

L447 to 454 (applies to corresponding results and discussion) I’m not convinced that this information is interesting. If the authors choose to keep it, they should justify its relevance here from a biological or conservation point of view. Personally, I would remove it to make the article

more concise. The point of song behavior attracting further bats to rotor-swept zones can still be made in the discussion.

Lines 487-488: We have added the following justification for calculating the signaling range of male songs. “Due to the attractive function of bat song^{25,44}, we wanted to quantify the distance over which males singing at wind turbines may be heard by conspecifics.”

We would like to keep this calculation in the manuscript. By doing so, we can not only discuss that males singing at wind turbines may attract further bats but can also make a quantitative estimate about the distance over which attraction may operate.

L462 “leading to a data rate of about 23 GB/hour” can be removed, as it is not necessary for replication.

Line 510: Done

L502 replace “not bias” by “avoid biasing”

Line 552: Done

L508 “spherical shells”

Line 560: Done

L509 I’m not sure I understand well, what do the authors mean by “straight forward”?

Lines 560-563: The three-dimensional volume of the shells was calculated using geometric principles, considering the cut-off of certain shells due to the viewing angle of the thermal cameras

L511 “We then calculated bat density by distance to the nacelle as the number of bat positions divided by spherical shell volume”.

Lines 562-563: Done

Figure 1 the main file contains quite a lot of figures, perhaps this first one can go to the supplementary material, since it is not really important for the Discussion section.

We agree and have moved the former Figure 1 to the Supporting Information.

Figures 2, 4 Reference to the natural regions of Germany may be removed to make the figures less crowded.

Figures 1 and 3: We’d prefer to keep the references to the natural regions of Germany in the figures. We are aware that its information value is quite limited for non-German readers but

would like to keep it in the figures for the large community of people from Germany interested in this topic.

Figure 3 “Pipistrelloid species showed proportionally higher social activity [...]” The authors could actually run a statistical test for this (e.g. a beta regression), just to be more confident in the significance of the difference for August.

Figure 2: Thank you for this suggestion. We have conducted additional statistical analyses using generalized linear mixed models (GLMMs) with a binomial error structure and logit link function to test for differences in both social and feeding activity at wind turbines between Pipistrelloid and Nyctaloid species. These models included species group, month, and their interaction as fixed effects, with random effects for site and observation to account for variability and overdispersion (see lines 459-474 of the methods section).

The results of these analyses confirmed that Pipistrelloid bats exhibited significantly higher probabilities of social activity across all months compared to Nyctaloid bats, with the differences being particularly pronounced in late summer and autumn. Specifically, post-hoc tests on the interaction terms demonstrated significant differences between the two species groups for social activity in September ($p = 0.0001$) and October ($p = 0.0013$). This supports the conclusion that Pipistrelloid species show proportionally higher social activity at wind turbines during late summer and autumn. Furthermore, these models also revealed that the differences in social activity between the two species groups were more pronounced than those for feeding activity. Given these results, we believe that the GLMM approach provides robust statistical evidence for the observed differences in social activity. We have updated the text in the Results (Lines 130-141) and Discussion (Lines 301-310) sections accordingly to reflect these new statistical analyses. In addition, we have added a new figure 2 (Estimated marginal means for feeding and social activity from the GLMMs) as well as a new Table 1 (GLMM results) and two new tables in the SI covering the results from post hoc tests (tables S3-S4).

Figure 4 The lower panel containing song excerpts can be moved to the supplementary material, and the upper panel can be combined with figure 2, reducing the total number of figures in the article by yet one less.

Thank you for your suggestion regarding former Figure 4 (now Fig. 3). As our study places particular emphasis on bat song at wind turbines, we find it important to retain the lower panel in the main text to highlight how bat songs appear acoustically. In addition, we feel, combining the upper panel with Figure 2 would overly complicate the figure and reduce clarity. Thus, we believe the current format supports our focus on bat song.

Figure 5 *N. leisleri*, *Pipistrellus* sp. and *Plecotus* sp. have very few observations. They could be left out, making the figure less crowded, and the results easier to interpret. In the caption, line 789, do the authors mean Table S5?

Now Figure 4: Thank you for your suggestion regarding former Figure 5. While we acknowledge that *N. leisleri*, *Pipistrellus* spp., and *Plecotus* spp. have relatively few observations, we find it important to include them to provide a complete representation of

the data. Excluding these species would not align with our aim to present the full spectrum of bat activity at wind turbines, even for species with limited data. We believe their inclusion, despite the smaller sample size, adds valuable context and ensures transparency in reporting our results. We have corrected the Table information, thanks for spotting this!

Figure 6 (panel B) the circle corresponding to the limit of the rotor-swept zone could perhaps be highlighted in another color. Also, in panel C, the limit of the collision-risk zone could be indicated with a vertical red line at the 35m distance to the nacelle position, for example.

Now Figure 5: We have added a blue circle (panels A and B) and blue vertical line (panel C) to indicate the limit of the collision-risk zone. Thank you very much for this suggestion.

Figure S2 Instead of enumerating the vocalization types in the caption, they could be labelled directly on the figure.

Now Figure S3: We have tried to implement this suggestion but ultimately decided against it because labelling the vocalization types directly in the figure made it very cluttered, especially because the label could not be set at uniform heights without interfering with the vocalization types.

Figure S5 Could the authors keep the same axis dimensions (not the scale) for panel E as the rest of panels?

Now figure S6: We have adjusted the panel dimensions for panel E.

Figures S6, S7, S8 What is the scale of the Y axis?

Now figures S7, S8, S9: We have added the information that the upper panels of these figures have a pseudo-logarithmic y-axis.

Figure S10 this figure is redundant with Figure 3. The authors could either show October in figure 3, or just remove this figure, since, as is mentioned in the caption, information on this month is not very interesting due to uncertainty in the mean.

Now figure S11: Figure 2 in the main manuscript now shows the estimated marginal means for social and feeding activity from the GLMMs. In the models, we have also included the data from October. Thus, now we only show the monthly proportions of bat call recordings containing feeding buzzes or social vocalizations in the supporting information. We have, however, adjusted figure S11 and removed the grey shading for August and September, since our GLMM results, indicated a significant difference in social activity between the species groups for September and October.

Figure S12 I'm not sure that the German landscape classification adds much information to a non-German reader. The map could be simplified removing these subdivisions.

Now figure S13: We agree, yet, would like to keep the German landscape classification.

Figure S13 Could the authors please describe the figure panels (A and B)?

Now Fig. S14: We have added panel descriptions to the figure legend:

(A) Two cameras were positioned 200 m from the wind turbine, with a baseline distance of approximately 16 m. Both cameras were aligned to capture the same reference point on top of the turbine's nacelle (orange dot), maximizing the overlapping field of view at the turbine's position. (B) The position of an unknown object (e.g., a bat) in space is determined using trigonometric calculations. Known distances, measured with a laser rangefinder, provide the angles α_1 and α_2 which are essential for solving the trigonometric equations. The angles β are derived from image analysis, considering the known focal length (i.e., the distance between the sensor and lens).

Figure S14 Same comment as for figure S13

Now Fig. S15: We have added panel descriptions to the figure legend:

“Multiple drone flights were conducted parallel to the deployment of the stereo thermal camera system. (A) Example of a three-dimensional flight path of a GPS-equipped drone. Drone flights were conducted to evaluate the localization accuracy of the stereo thermal camera system. (B) Comparison of the two-dimensional distance between the drone and the nacelle, based on GPS data (blue line) and stereo thermal triangulation results (golden line). (C) Comparison of the three-dimensional distance between the drone and the nacelle, using GPS data (blue line) and stereo thermal triangulation results (golden line). No systematic errors were detected. Mismatches illustrated by non-overlapping sections of the blue and gold lines resulted from the accuracy of the GPS and two-dimensional detection error in determining the drone's center.”

Table S2 This table might be more compact and easier to read if the species are placed as columns, avoiding the repetition of syllable/song element types.

Table S2: Many of the syllable types are unique to species, do not repeat and may not exist in other species. We have, thus, chosen the current format of the table to avoid giving the impression that any of these syllable types or song element types are present in a species but were not recorded.

Table S4 As in figure 5, data for *N. leisleri* and *Plecotus* sp are insufficient to be considered representative, and can be left out.

Now table S6: We find it important to include them to provide a complete representation of the data. We report their small sample sizes in the table. We believe their inclusion, despite the smaller sample size, adds valuable context and ensures transparency in reporting our results.

Table S5 A and B are quite separate tables, they may be numbered separately.

Now Tables S7 and S8. Done

Table S9 caption, lines 195 – 197 (“Therefore [...] nacelle”). This information is already given in the Methods section, so this part of the caption may be removed.

Now table S12: Thank you for the suggestion. We believe it is useful to retain the mentioned part of the caption for clarity and ease of reference.

Reviewer #2 (Remarks to the Author):

This manuscript came as a shock to me. Having been deeply involved in the initial response and scientific investigation into bat fatalities at wind turbines in North America, about a decade ago I retreated from active research into the underlying causes of this novel and deeply complex problem.

I withdrew in response to growing obstacles to testing attraction hypotheses and frustration that North American researchers and their ubiquitous industry and/or government regulatory partners seemed more focused on engineering solutions to the bat-turbine problem than investigating its root causes and better understanding the behaviors and natural histories of the bats in question.

Research into machine-based engineering solutions and turbine operational changes has been consistently prioritized in the U.S. and Canada over basic biological research and what I consider a life-history perspective. As a result, we still lack clear and evidence-based scientific explanations for why the proportionally few species of bats most affected by wind turbines in the U.S. and Canada (some of which may go extinct because of wind turbines) seem attracted to the very structures we are working so hard to keep them away from with operational changes and new mechanical inventions. That could all change with the publication of this study.

The elephant in the room, which the authors diplomatically refrain from bringing up in this exemplary manuscript, is that those of us in the scientific weeds of the problem in other parts of the world lag far, far behind in having the means to test, let alone carry out, meaningful observational studies of bats around operational wind turbines. The authors seem to have taken the opportunities and information available to them and done everything right. This paper will undoubtedly be a surprising game changer with global impact.

I've never used such a glowing adjective in my 30+ years of reviewing scientific papers, but I view this manuscript and study as a masterpiece. Without hyperbole, it reports findings based on clear and sufficient evidence of a nature that I never expected to see during my scientific career. I knew that the authors had long been deploying good bat detectors on wind turbines across Germany; I knew that there was a rich background of natural history observations and understanding of European bat behaviors that went far beyond the usual focus on those occurring in nice weather, close to the ground, and in beautiful natural areas; and I knew that there was more focus on investigating the underlying causes of bat fatalities at wind turbines in Europe than other regions. I did not know the methods and technologies for recording, analyzing, calibrating, bias correcting, validating, and interpreting these rare observational events had been so carefully and meticulously developed, tested, and implemented. I found no holes in the assumptions, reasoning, data analyses, validation methods, or inferences drawn despite reading the manuscript carefully twice and skimming it a couple times more.

The writing is clear and concise, the figures are beautiful, and the legends provide sufficient information for the reader to easily grasp their meaning at a glance while drawing deeper understanding with more careful reading. The various analytical frameworks used for

interpreting the 2D and 3D spatial distributions of sound and visual signals are creative and more easily repeatable by future investigators than any I have yet seen. In general, all the methods seem easily repeatable for future studies given the level of detail provided. The illustrations of feeding buzzes, social call varieties, and songs among several species will undoubtedly serve to educate and surprise many readers and scientists, including bat researchers, who were ignorant of their existence. I think the authors essentially 'write the book' about how this kind of study can be done and then reported very successfully. Even the supporting information impressed me. For example, Table S9 showing the inferred differences between optical flight observations and acoustic detections of bats around turbines is something I have personally attempted and completely failed at several times, despite great cost and effort. So subtle a table--such valuable information.

Dear Reviewer 2:

We are truly overwhelmed and deeply honored by your incredibly generous and thoughtful review. Hearing this from someone with such extensive experience and historical perspective on the challenges of understanding bat interactions with wind turbines is especially meaningful. After facing numerous desk rejections for this manuscript, your words feel like a profound recognition of our work. We sincerely appreciate your time, expertise, and kind words.

Aside from a few rare editorial glitches (see detailed comments that follow), the only substantive suggestion I have for improving the text is to add a bit of detail about how the thermal cameras were temporally synchronized for the 3D analysis. I'm guessing they were tethered to the same capture device or that their clocks were somehow regularly synchronized? I also wondered if the drone was supplementally heated, or if its regular operational waste heat was sufficient to detect with the thermal cameras...very minor questions...and few.

Thank you very much for this suggestion. We have added information to the methods section concerning thermal camera synchronization. We now write on lines 525-527: "Both thermal cameras were linked to a single frame grabber, which was operated via LabVIEW to ensure synchronized frame acquisition."

On lines 540-541 we also added the following information: "The drone did not require additional heating, as the heat generated by the rotors and electronics was sufficient for localization with the thermal cameras."

Another suggestion I am still on the fence about making is whether to recommend going a bit further out into speculation in the discussion section. The lack of physical evidence for song flight in the North American species of bats most affected by wind turbines (*Lasiurus cinereus*, *L. borealis*, *Lasionycteris noctivagans*) isn't likely due to it not occurring, but because few people have ever looked for it and, if found, characterized social calls and songs in these species of bats. Three studies that stand out in this regard that were not cited hint at the possibility that something similarly complex might be happening in North America. However, we lack the background natural history observations and have thus far missed ample research opportunities to uncover such connections. These papers were:

Reyes, G.A., Szewczak, J.M. Attraction to conspecific social-calls in a migratory, solitary, foliage-roosting bat (*Lasiurus cinereus*). *Sci Rep* 12, 9519 (2022). <https://doi.org/10.1038/s41598-022-13645-9>

Corcoran, A.J., Weller, T.J., Hopkins, A. et al. Silence and reduced echolocation during flight are associated with social behaviors in male hoary bats (*Lasiurus cinereus*). *Sci Rep* 11, 18637 (2021). <https://doi.org/10.1038/s41598-021-97628-2>

Cryan P.M., Jameson, J.W., Baerwald, E.F., Willis, C.K.R., Barclay, R.M.R., Snider, E.A., et al. Evidence of late-summer mating readiness and early sexual maturation in migratory tree-roosting bats found dead at wind turbines. *PLoS ONE* 7(10): e47586 (2012). <https://doi.org/10.1371/journal.pone.0047586>

These scrappy observations from elsewhere may or may not be worth bringing up in the discussion, since any well-informed readers that see this paper in North America will grasp the relevance and possible connections. However, those of us in North America would welcome some advice about how to go about looking into these questions by those who succeeded in doing so.

Many thanks for this suggestion. We have incorporated a small paragraph into our discussion that suggests that social and mating related behaviors may also contribute to the attraction of North American bats to wind turbines (Lines 260-268).

“To date, there is evidence of song produced in flight in only one (*Lasionycteris noctivagans*) of the three North American migratory bat species (*Lasiurus cinereus*, *L. borealis*, *L. noctivagans*) most affected by wind turbines⁴⁷. However, recent studies indicate that *L. cinereus* use and are attracted to social calls during migration^{48,49}. Additionally, in one study most bats of these species found dead below wind turbines exhibited physiological signs of mating readiness⁵⁰, suggesting that social and mating related behaviors may contribute to their attraction to wind turbines²⁰. This underscores the importance of further research into the mating systems and behaviors of these North American species, including mating related social communication and the potential use of song flight to attract mating partners.”

Another thing I wondered about while pouring over the figures and tables about the seasonal prevalence of song flight and social behaviors in the Pipistrelloid and Nycaloid groups was whether the authors think the patchiness of social behaviors observed among sites and thus regions might have to do with migratory tendencies of certain group members? For example, does the higher prevalence of Pipistrelloid social calls and songs at turbines in August through October reflect some species of Nyctaloids being more likely to leave Germany in late summer for mating and wintering areas elsewhere? It seems that being in the right place at the right time, or vice versa, might further help explain those broad seasonal pattern differences among feeding and social detections? As my poorly articulated question indicates, this may not be an easy question to address without a lot of text, but I'm bringing it up here in case there might be a concise way of touching upon it.

Thank you for this excellent suggestion. We have added a sentence to the discussion proposing that the differences in social activity between these two groups may be influenced by variation in migratory behavior and/or the relatively sedentary nature of some pipistrelles.

Lines 307-310 “The particularly large differences in social activity between Nyctaloid and Pipistrelloid species groups at wind energy facilities in September and October could potentially be explained by differences in migration timing, as well as by the fact that two species within the Pipistrelloid group are more sedentary (e.g., ^{58, 59}).”

My minor editorial notes:

Line 177: is “spec” the journal standard for species or is it spp?

Line 120 and throughout the main Manuscript and SI: We believe the reviewer is right that the journal standard is spp. and have changed spec and sp. to spp..

Line 209: is the plural of nacelle nacelles?

Line 202: yes, the plural is “nacelles” and we changed it accordingly.

Line 245: “...or the edge[s] of woodlands...”

Line 239: done

Lines 256-257: “...song flight is performed in [close proximity to] mating roosts and flight paths also often follow [linear or prominent landscape features] such as [forest edges] or buildings...”

Lines 250-251: Done

Line 315: “...might help [explain] the highly female-biased fatality ratios.”

Line 325: Done

Line 360: is this time in UTC?

Line 375: it is UTC + 02:00, which is the Central European Summer Time. We have added this information to the manuscript.

Line 371: maybe add a sentence about what the ‘BMU-settings’ are for readers without access to the software or documentation?

Line 387 ff: We have added specific information on these settings. Please also note that in the new software version of the RECORDER software these settings have been renamed to ‘RENEBAT settings’ and we are now also using this new term in our paper.

Line 421: comma after i.e.?

Line 445: Done

Line 445: "...species-specific bat song, [we inferred that] singing bats were not simply passing..."

Line 485: Done

Line 511: hyphenate? distance-dependent bat density

Lines 562-563: We modified this sentence based on a suggestion from reviewer 1 and, thus the term distance-dependent has been removed from the sentence.

Page 14, Line 123: "Wind turbine sites were sample[d] in four different..."

Page 17, Line 127: Done

Page 14, Lines 125-126: hyphenate so-called?

Page 17, Line 129: Done

Page 16, Lines 137-138: "No systematic errors were detected[. Mismatches illustrated by non-overlapping sections of the blue and gold lines resulted] from the accuracy of the GPS [and] two-dimensional detection [error in determining] the drone's center."

Page 19, Lines 151-153. Done

Reviewer #3 (Remarks to the Author):

The manuscript is well written adds to the growing body of knowledge regarding bat interactions with wind turbines. Below are a few minor suggestions and a few moderate to major suggestions to improve the manuscript.

Dear Reviewer 3:

We would like to express our gratitude for the positive feedback and excellent suggestions. In our revision, we have carefully refined our conclusions to ensure greater clarity. Additionally, we recognized that our definition of attraction was not sufficiently clear. To address this, we have added a sentence in the introduction (lines 57–59) clarifying the distinction between long-distance attraction to wind turbines and an increase in local attractiveness. While it is possible that males may be drawn to wind turbines from greater distances due to certain turbine traits (lines 244–254), our primary argument is that if males sing at wind turbines, this behavior enhances the site's local attractiveness for bats already in the vicinity. This, in turn, may lead to prolonged presence in the collision risk zone. Since males engage in song flight, their vocalizations can be projected over a larger area, potentially attracting even more bats (lines 239-243).

Line 28: There are more recent syntheses of curtailment studies available. See Adams et al. 2021 'A review of the effectiveness of operational curtailment for reducing bat fatalities at terrestrial wind farms in North America' and Whitby et al. 2021 'The state of the science on operational minimization to reduce bat fatality at wind energy facilities.

Line 31: We have added a citation for Whitby et al. 2021.

Line 32-33: The reports on the 'cost' of curtailment are in terms of annual energy production and not revenue. See Whitby et al. 2022 for more information on the loss of annual energy production from curtailment in the US.

Lines 36-40: Thank you for pointing that out. We have rephrased this part and now cite Whitby et al. 2021 for the “costs of curtailment” in North America.

“In North America, for example, loss in annual energy production has been estimated to be mostly below 1 % and up to 3.2% for curtailment during the main collision risk periods of the year¹³. Similarly, a German study that reduced bat mortality to two dead bats per year with turbine specific curtailment algorithms found that yearly revenue decreased by 1 – 2 %¹⁰. Costs may be higher for modern turbines, but published data on this are lacking.”

Line 41 and line 52: Suggest adding Guest et al. 2022. 'An updated review of hypotheses regarding bat attraction to wind turbines'.

Lines 46 and 57: Done

Line 54: There is a more recent analysis on the relationship between pre-construction activity and post-construction fatality. See Solick et al. 2020 'Bat activity rates do not predict bat fatality rates at wind energy facilities'.

Line 62: We are now citing the more recent publication.

Relating to section 'Bat songs at wind turbines' starting on line 137: 1) How many nights were social calls recorded? How many social calls/night? On nights with social calls, was there an increase in bat activity relative to non-social call nights (this would help make the connection that social calls are attracting bats)? How does the rate of social call activity at wind turbines relate to airspace without wind turbines (does it occur more often or less often)? I realize your study may not be able to answer the last one because of how the study was set up, but if there is no difference in social call activity from wind turbines to other areas then it cant be considered an attractant.

Thank you for the thoughtful comments. Regarding social vocalizations and songs, we recorded social vocalizations on 313 turbine nights, averaging 12 ± 7.4 nights per turbine (8.5% of nights per turbine) and songs on 81 turbine nights, averaging 3.1 ± 2.9 nights per turbine (1.9% of nights per turbine). These results indicate that social activity at wind turbines is not an isolated occurrence but is instead a regular component of bat behavior in these environments.

We are not convinced that testing whether bat activity is higher on nights with social calls is indicative of bat attraction. Social calls are correlated with overall bat presence (number of echolocation recordings), as shown in former Figure 1 (now Supporting Figure S1). We interpret the data from Figure S1 that the presence of more bats at wind turbines simply means more social interactions. Thus, we believe that our data already show that social activity increases with overall bat activity at wind turbines, but that it is probably not indicative of attraction.

As for comparing social call rates between turbines and non-turbine areas, we unfortunately do not have such data.

Lind 235: 'considerable distances'. As stated above, 42 m and 100 m are not considerable distances. To hear these calls, bats would already have to be within close proximity to the turbine.

Line 228-229: We have modified this sentence to : “ ... *over much higher distances than echolocation calls* ... “. Bats also perform song flight to increase the range of their songs. We agree with the reviewer that male songs are unlikely to draw bats from afar to wind turbines. However, we believe it is plausible that male song flights contribute to increasing the local attractiveness of turbine sites for other bats. To clarify this, we have revised the introduction to explicitly distinguish between long-distance attraction to wind turbines and increased local attractiveness. (see changes on lines 75-77 and lines 90-92).

Line 247: 'frequently'. Are these calls frequent? They dont appear to be based on the data presented.

Line 240: We have removed the term “frequently”.

Line 266: '...are as yet the strongest evidence for bat attraction'. I disagree that it is the strongest. It is additional data to a growing body of knowledge that points to certain species of bats being attracted to wind turbines/spending more

Lines 269-270: We have modified this sentence to: “Our results on bat density at nacelle height are strong evidence for bat attraction to wind turbines ... “.

Line 190: distance of social calls = 42 m for *Plecotus* spp. and 100 m for *P. nathusii*/ *V. Murinus*. These are not long distances. For *Plecotus*, another bat hearing the call would already have to be near the wind turbine already. The turbines have an approx 35 m blade length, so if a male *Plecotus* is singing at the nacelle, a listening bat would be only 7 m away from the tip of the blade. Similarly, for *P. nathusii*/*V. Murinus*, though the distance is longer, its still relatively short. This really eliminates any idea that bats are being drawn into the area by hearing calls. They are already close by. Now, it may increase the flight times of bats in and around wind turbines, thus increasing their risk. But its not a viable cause of attraction.

This calculation is only valid for a stationary bat. We assume that bats are flying while they sing, and this behavior probably increases the range/ active space of their songs considerably. We agree with the reviewer that male songs are unlikely to draw bats from afar to wind turbines, but male songs have an attracting effect on bats in hearing range and may draw those bats into the collision risk zone. Thus, we believe it is plausible that male song flights contribute to increasing the local attractiveness of turbine sites and may make these sites more appealing to bats already in the vicinity or commuting through the area and draw these bats into the collision risk zone. To clarify this, we have revised the introduction to explicitly distinguish between long-distance attraction to wind turbines and increased local attractiveness. (see changes on lines 75-77 and lines 89-92).

Line 214-215: 'Bat song was documented at all studied sites and in 88% of the 26 wind turbine-years, suggesting that it is not a rare behavior, but that bats commonly sing at wind turbines in Germany.' It certainly seems like a rare behavior. What is the percentage of calls of singing. If I understand it correctly out of approximately 83,000 recordings, 4,100 were social vocalizations, and 1,300 were songs. That doesnt seem like a common occurrence. For example, say the night is 8 hours long, and a song is 1 minute and there is only 1 song per night, that leaves 539 minutes of time when songs are not occurring/being heard. So, other bats would have to be within hearing distance at the exact 1 minute interval across the entire night to be attracted to it.

Lines 208-209: We agree with the reviewer that our wording was misleading here. We have rephrased this sentence to clarify that this was not a frequent behaviour, but that we did encounter it consistently across the studied turbines. “Bat song was documented at all studied sites and in 88% of the 26 wind turbine-years, suggesting that while it is not very common (92 song events in total), it is a consistently observed and recurrent behavior at wind turbines in Germany.”

Lines 278-281: '...it appears that evidence in favor of the feeding attraction hypothesis is increasing'. This makes it sound like the feeding hypothesis is out competing other hypotheses. Rather, it just points to the conditions that occur across your sample turbines. At one point it is mentioned that the turbines are on open crop land. That would be primary foraging habitat for bats regardless of the presence of wind turbines. So, to say that bats are foraging more around wind turbines in this landscape relative to other airspace, you'd need to have detectors at turbine and non-turbine locations. It would also be helpful to determine whether there are more insects in and around turbines relative to open airspace. Moreover, the surrounding habitat likely plays a role in how bats perceive wind turbines. An agricultural setting with new tall structures could present new roosting opportunities to bats as compared to a forested habitat. Alternatively, a forest habitat with new clearings for turbines and roads, may create new edge habitat for foraging and commuting. All this is to say that attraction hypotheses are not mutually exclusive, they are likely occurring at different scales, differ by species, differ by habitat, and possibly interacting in ways that we still don't fully understand.

Thank you for this suggestion. We agree and have changed the wording. On lines 283ff we now write: "Our results show that foraging activity occurred at all studied wind turbines with feeding buzzes from 2 – 6 species (-groups) detected per wind turbine year and may provide additional support for the feeding attraction hypothesis."

Figure 3: This speaks to how rare the feeding and social calls are. At no point to feeding buzzes exceed 5% even at the higher 95%CI. Similarly for social vocalizations (though in August the higher 95% gets slightly above 5%). Social vocalizations for Nyctaloid are near zero for 3 of the 5 months. August and September show the highest percentage of social vocalizations for Pipistrelloid. You would expect to see an increase in overall Pipistrelloid activity associated with these data if social vocalizations are attracting bats within this group.

We agree with the reviewer that neither feeding nor social activity were highly frequent at the studied wind turbines. We have rephrased the parts of the discussion, which suggested that this was a frequent behavior. We would, however, like to emphasize that we encountered these behaviors consistently and recurrently at our studied wind turbines.

Lines 207-209: Bat song was documented at all studied sites and in 88 % of the 26 wind turbine-years, suggesting that while it is not highly frequent (92 song events in total), it is a consistently observed and recurrent behavior at wind turbines in Germany.

Lines 283-285: Our results show that foraging activity occurred at all studied wind turbines with feeding buzzes from 2 – 6 species (-groups) detected per wind turbine year and may provide additional support for the feeding attraction hypothesis.

We believe that it should be clearer now that the two lines of evidence providing the strongest support for bat attraction to wind turbines in our manuscript are the 3D thermal imaging results on the density distribution of bats around the nacelle and the increase in local attractiveness of wind turbines due to the broadcasting of song by males and their performing of song flight.

Figure 6, cameras set up and sample size: Its difficult to make any definitive statements about the results from 6 nights of data collection across 4 wind turbines. That is a rather small sample. Moreover, the field of view is heavily skewed toward detecting bats close to the wind turbine

because the overlap of the 2 cameras is centered on the nacelle. If the field of view covered the entire 60 m concentric circles, that would be one thing, but Fig. 6 shows that most of the farther distances are not even in the field of view. The cameras don't even cover the entire rotor-swept area. It's also not surprising that you are seeing fewer observations the farther away from the camera as detection decreases with distance.

We acknowledge that our dataset, covering six nights across four wind turbines, is limited in size. However, it represents the best available 3D thermal imaging dataset on bat density at nacelle height to date. Despite the sample size, the consistency of our results across multiple sites, combined with alignment to previous studies, provides strong initial evidence for bat attraction to wind turbines. While additional research with larger datasets is necessary to further validate these findings, our study offers a crucial first step in quantifying bat behavior in the rotor-swept area. We have added information in the discussion to acknowledge that our data set is limited on lines 272-274: "While our dataset is limited in size, it provides the best available evidence for bat attraction to wind turbines at nacelle height. Future studies with larger sample sizes will be essential to further validate these findings."

Thank you also for your valuable comment regarding the spatial limitation of the camera's field of view and its potential impact on our results. To ensure the robustness of our analyses, we performed separate analyses of bat activity in front of and behind the wind turbines, as shown in the new Figure S18.

We have added the following text to the Methods section on lines 564-570: "We verified the robustness of our bat density calculation through the separate analysis of different spatial areas (see fig. S18). Although the cameras did not cover the entire rotor-swept area, comparable exponential decreases in bat density with increasing distance from the nacelle were observed in front as well as behind the wind turbine. The lower bat density behind the wind turbine can be explained by a general reduction in detection probability with greater distance from the cameras and by lower activity of all bat species at higher altitudes^{55,84}. The area behind the wind turbine covered by the cameras was higher above ground than the area in front of the turbine."

While it is true that the cameras did not cover the entire rotor-swept area, we observed a comparable exponential decrease in bat activity density across the analyzed subsets. This consistency suggests that the density estimates of bats at the nacelle derived from the data are robust despite the limitations in the camera's field of view.

RESPONSE TO REVIEWERS' COMMENTS:

Reviewer #1 (Remarks to the Author):

Dear Authors, Dear Editor,

Thank you for your patience in waiting for my report. I find the manuscript has gained in clarity concerning its main claims and the supporting evidence. I'd also like to commend the authors for their efforts to provide all the necessary details for study reproducibility, which is something we often lack.

I personally would've preferred to see a results section that avoids giving methods information (e.g. lines 143-151 and 173-179) and rather synthetically presents the evidence. Also, the paragraph in lines 260-268 is not very well woven into the discussion. Still, these points are not really important, and it is above all desirable to see this article published soon so that the community can finally have access to it. I find no major issues to be reported, and am happy to recommend its publication in Communications Biology.

Response to Reviewer #1:

We would like to express our gratitude for the positive evaluation of our manuscript and for the constructive suggestions. We appreciate the reviewer's suggestions concerning the structure of the Results and Discussion sections. However, we have intentionally included a certain level of redundancy to enhance clarity and transparency. Therefore, in line with the editor's guidance that no further revisions are mandatory, we have not restructured these sections. We are grateful for the reviewer's support and recommendation for publication.

Reviewer #3 (Remarks to the Author):

The authors did a great job of addressing my comments. I really appreciated their thoughtful responses and how their efforts to revise the manuscript. I have no further comments. Great job.

Response to Reviewer #3

We would like to express our gratitude for the positive evaluation of our manuscript!